# Argyrodite-type advanced lithium conductors and transport mechanisms beyond paddle-wheel effect

Hong Fang [1✉] & Puru Jena [1✉]

Development of next-generation solid-state Li-ion batteries requires not only electrolytes with high room-temperature (RT) ionic conductivities but also a fundamental understanding of the ionic transport in solids. In spite of considerable work, only a few lithium conductors are known with the highest RT ionic conductivities ~ 0.01 S/cm and the lowest activation energies ~0.2 eV. New design strategy and novel ionic conduction mechanism are needed to expand the pool of high-performance lithium conductors as well as achieve even higher RT ionic conductivities. Here, we theoretically show that lithium conductors with RT ionic conductivity over 0.1 S/cm and low activation energies ~ 0.1 eV can be achieved by incorporating cluster-dynamics into an argyrodite structure. The extraordinary superionic metrics are supported by conduction mechanism characterized as a relay between local and long-range ionic diffusions, as well as correlational dynamics beyond the paddle-wheel effect.

[1] Department of Physics, Virginia Commonwealth University, Richmond, VA, USA. ✉email: hfang@vcu.edu; pjena@vcu.edu

Given the world-wide concern about global warming and the urgent need to transition from fossil fuels to green energy, lithium-ion batteries will continue to be an integral part of our lives. Development of the next-generation solid-state lithium batteries that are safe and powerful relies on the lithium superionic conductors with high room-temperature (RT) ionic conductivities and low activation energies. To this date, the highest RT ionic conductivity ($\sigma_{RT}$) of the lithium superionic conductors is in the order of $10^{-3}$–$10^{-2}$ S/cm (1–10 mS/cm) with activation energies ($E_a$) of 0.2–0.3 eV. Belonging to only a handful of material classes, these include lithium sulfides, e.g., $Li_{10}GeP_2S_{12}$ (LGPS) ($\sigma_{RT} = 12$ mS/cm, $E_a \geq 0.25$ eV)[1] and $Li_2S$-$P_2S_5$ systems ($\sigma_{RT} = 3$ mS/cm, $E_a = \sim 0.2$ eV)[2], garnets, e.g., $Li_7La_3Zr_2O_{12}$ (LLZO) ($\sigma_{RT} = \sim 1$–2 mS/cm, $E_a = \sim 0.3$ eV)[3], NASICON-type $Li_{1.3}Al_{0.3}Ti_{1.7}(PO_4)_3$ (LATP) ($\sigma_{RT} = \sim 1$ mS/cm, $E_a = \sim 0.3$ eV)[4], lithium-rich antiperovskites, e.g., $Li_3OCl_{0.5}Br_{0.5}$ ($\sigma_{RT} = \sim 1$–2 mS/cm, $E_a = \sim 0.2$–0.3 eV)[5] and lithium argyrodites, e.g., $Li_{6.72}PS_5Cl$ ($\sigma_{RT} = \sim 1$ mS/cm, $E_a > 0.2$ eV)[6,7]. In spite of significant progress made in optimizing the lithium conductors over the past two decades and recent computational studies predicting many more promising lithium conductors through chemical and defect engineering (e.g., element substitution, doping, and vacancy-defect introduction)[8–11], no lithium conductor so far can reach or even come close to the criteria of the so-called advanced superionic conductors (ASIC)[12] with $\sigma_{RT} > 100$ mS/cm and $E_a \sim 0.1$ eV that were found for mobile $Ag^+$ and $Cu^+$ in the rubidium silver iodide family (e.g., $RbAg_4I_5$) more than four decades ago[13,14].

Achieving lithium conductors with ultra-high RT ionic conductivities and extremely low activation energies requires a fundamental understanding of the superionic behavior of Li-ions in the solids. Early generic theory found that the conditions that are responsible for high ionic conductivity are realized with monocation in a four- (tetrahedral) or three-coordinated configuration with the anions[15]. Recent studies have furthered this idea to link the high ionic conductivities observed in lithium sulfides (e.g., LGPS and $Li_7P_3S_{11}$) to body centered cubic (bcc) sulfur-sublattice which can lead to energetically-equivalent tetrahedral (four-coordinated) sites of lithium and large transport channels[16]. Other studies have found that collective lithium transport can flatten the migration barrier and promote the fast-ion conduction in LGPS, cubic LLZO and LATP, where two to three Li-ions on average migrate together[17–20].

Most recently, superionic conductors (such as $Li_3PS_4$, $Li_2B_{12}H_{12}$, $NaCB_9H_{10}$, $LiNaSO_4$, $Na_2SO_4$, $Na_3PO_4$, $LiBH_4$, $Li_3SBF_4$, $Na_3SBCl_4$, etc.) containing anion clusters (i.e., $PS_4^{3-}$, $B_{12}H_{12}^{2-}$, $CB_9H_{10}^-$, $SO_4^{2-}$, $PO_4^{3-}$, $BH_4^-$, $BF_4^-$, $BCl_4^-$, etc.) have become a fertile ground to search for high ionic conductivities[21–30]. An interesting feature of these ionic conductors is the so-called paddle-wheel effect characterized by strong correlation between the translational motion of the cation and the rotation of the cluster[21]. Under thermal excitation, non-spherical clusters can lead to a rotationally disordered phase, which can create low-energy pathways for ionic migrations and greatly enhance the ionic conductivity[10,22,23,31]. This is exemplified by the high Na-ion conductivity of 30 mS/cm of $NaCB_9H_{10}$ as it enters a rotational disordered phase at room temperature. Therefore, fully exploiting the rotational degrees of freedom and utilizing the paddle-wheel effect could be a viable way to achieve lithium conductors with near-ASIC metrics. However, other than $NaCB_9H_{10}$, most of the known cluster-containing ionic conductors can only transition to the rotational disordered phase at high temperatures[32–34]. It has been found that an amorphous structure with a low density and a relatively large volume to accommodate the cluster can support the paddle-wheel effect even at low temperatures (e.g., room temperature)[21,31]. Yet, the rotational freedom of the clusters in such a system seems to be limited,

as it is still not comparable to that of the high-temperature phase of $Li_2B_{12}H_{12}$[21,23]. On the other hand, one notices that the high charge states of the clusters (e.g., '-3' of $PS_4^{3-}$ and '-2' of $B_{12}H_{12}^{2-}$) and their large moment of inertia may increase their local interactions and hinder their rotations under thermal excitation. Therefore, a combination of light mono-anion clusters with a chemically/structurally accommodating lattice framework could be the key to achieve exceptional lithium conductors that enjoy great rotational degrees of freedom of cluster at room temperature.

Here, by incorporating light mono-anion clusters, $SH^-$ and $BH_4^-$, into an argyrodite framework, we discovered, based on first principles theory, two advanced lithium superionic conductors (ALiSIC)—$Li_6POS_4(SH)$ and $Li_{6.25}PS_{5.25}(BH_4)_{0.75}$ with $\sigma_{RT} = 82$ mS/cm ($E_a = 0.166$ eV) and $\sigma_{RT} = 177$ mS/cm ($E_a = 0.108$ eV), respectively. In these materials, $SH^-$ and $BH_4^-$ exhibit extremely high rotational as well as translational degrees of freedom. For example, at room temperature, $SH^-$ is found to rotate up to 180° within 9 ps and $BH_4^-$ can rotate up to 180° within just 2 ps. Both clusters exhibit translational vibrations close to 1 Å which is two orders of magnitude higher than the normal thermal vibrational amplitude ($\sim 0.01$ Å) of an atom. Through a systematic study of the Li-ion transport together with the cluster dynamics in these systems, a complete physical picture of ionic conduction is established. Mechanisms and correlational dynamics beyond the paddle-wheel effect are uncovered to explain the extraordinary fast-ion diffusion, especially, a mechanism that enables coupling between the local and long-range diffusions.

## Results

We use the chemical composition and structure of lithium argyrodite family ($Li_6PS_5X$, $X = Cl$, Br, I)[7] as a template and substitute the halogen atoms (X) by the clusters (such as $OH^-$ and $BH_4^-$) that are readily available. These clusters, known as super- or pseudo-halogens, have comparable sizes with halogens and can partially mimic their chemistry[10,35]. The structures of the newly-developed cluster-based materials are obtained from extensive first-principles structure search (see Method). When $OH^-$ is introduced to make a compound $Li_6PS_5(OH)$, the ground state structure turns out to be $Li_6POS_4(SH)$ containing $SH^-$ and $POS_3^{3-}$ instead of $OH^-$ and $PS_4^{3-}$ clusters, as shown in Fig. 1A. The optimized ground-state structure of $Li_6PS_5(BH_4)$ is shown in Fig. 1B.

The calculated phonon dispersions confirm that the triclinic phases of these materials are lattice-dynamically stable (Supplementary Fig. 2). Note that the lattice-dynamically stable ground states of the argyrodites with halogens are also triclinic instead of cubic[11]. The calculated band gaps of $Li_6POS_4(SH)$ and $Li_6PS_5(BH_4)$ are 4.37 and 3.47 eV, respectively (Supplementary Fig. 3), which are greater than those of the halogen argyrodites ($\leq 3.40$ eV). The calculated formation energies, corresponding to the reaction $Li_3PS_4 + Li_2S + LiY = Li_6PS_5Y$ (Y = the cluster or halogen), of $Li_6POS_4(SH)$ and $Li_6PS_5(BH_4)$ are 16 and 18 meV/atom, respectively. These are significantly smaller than that of the triclinic (stable) $Li_6PS_5Cl$ (29 meV/atom).

Based on the molecular dynamics (MD) simulations, the calculated RT ionic conductivities of $Li_6POS_4(SH)$ and $Li_6PS_5(BH_4)$ are 82 and 0.2 mS/cm with activation energies of 0.166 and 0.333 eV, respectively (see Method). It is known that the non-stoichiometric configuration of the lithium argyrodite, $Li_{6.25}PS_{5.25}Cl_{0.75}$, exhibits a RT ionic conductivity of 14 mS/cm which is more than four orders of magnitude higher than the $2 \times 10^{-3}$ mS/cm of the stoichiometric $Li_6PS_5Cl$[8]. $Li_{6.25}PS_{5.25}Cl_{0.75}$ also has an activation energy of 0.21 eV which is much smaller than that (0.52 eV) of $Li_6PS_5Cl$[8]. Similarly, we find that the

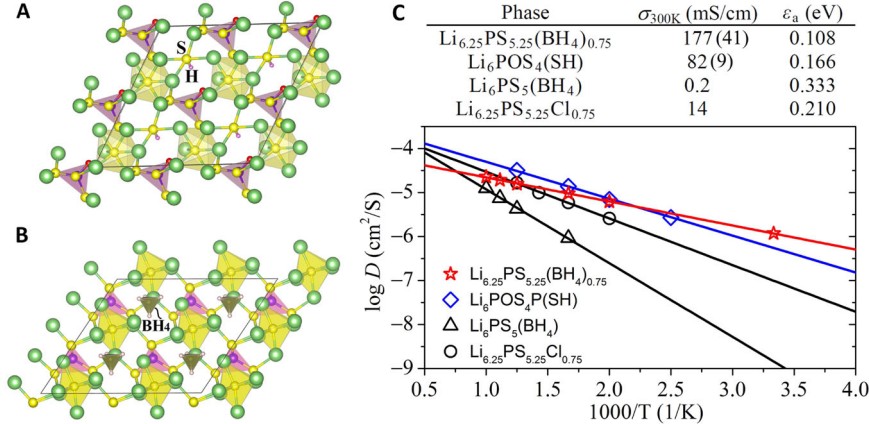

**Fig. 1 The predicted lithium superionic conductors based on the chemical composition of the argyrodite family, $Li_6PS_5X$ (X=halogen), where X is replaced by cluster-ions $OH^-$ and $BH_4^-$. A** The optimized $2 \times 2 \times 2$ ground-state structure of $Li_6POS_4(SH)$ (Triclinic: $a = 5.9662$, $b = 6.7378$, $c = 6.9147$ Å; $\alpha = 117.3107°$, $\beta = 88.072°$, $\gamma = 90.4551°$) containing polyatomic units of rod-like $SH^-$ (S in yellow and H in pink) and tetrahedral $POS_3^{3-}$ (P in violet and O in red). The optimized $Li_6PS_5(OH)$ configuration (Supplementary Fig. 1) containing $OH^-$ and $PS_4^{3-}$ cluster units is 140 meV/formula higher in energy. **B** The optimized $2 \times 2 \times 2$ ground-state structure of $Li_6PS_5(BH_4)$ (Triclinic: $a = 6.9678$, $b = 7.016$, $c = 7.0769$ Å; $\alpha = 61.9646°$, $\beta = 117.5361°$, $\gamma = 118.4223°$) containing polyatomic units of tetrahedral $BH_4^-$ (B in dark green and H in pink) and $PS_4^{3-}$ (P in violet and S in yellow). In both $Li_6PS_5(OH)$ and $Li_6PS_5(BH_4)$ structures, every group of six Li atoms (in green) form a distorted octahedron centered at sulfur. **C** Calculated RT ionic conductivities ($\sigma_{300K}$) and activation energies ($\varepsilon_a$) in table for nonstoichiometric $Li_{6.25}PS_{5.25}(BH_4)_{0.75}$, $Li_6POS_4(SH)$, $Li_6PS_5(BH_4)$ and nonstoichiometric $Li_{6.25}PS_{5.25}Cl_{0.75}$. The uncertainties (in the parentheses) of the $\sigma_{300K}$ for $Li_{6.25}PS_{5.25}(BH_4)_{0.75}$ and $Li_6POS_4(SH)$ are estimated based on the statistical variance analysis introduced in[36]. The plot shows, for each material, the calculated diffusivities at different temperatures fitted by the Arrhenius relation (see Method). The data points in each case show excellent linearity. The shaded data points correspond to the simulation temperature of 800 K at which the probability distribution function in Fig. 2 is calculated.

nonstoichiometric $Li_{6.25}PS_{5.25}(BH_4)_{0.75}$ exhibits a RT ionic conductivity of 177 mS/cm—three orders of magnitude higher than that of its stoichiometric counterpart $Li_6PS_5(BH_4)$ and one order of magnitude higher than that of $Li_{6.25}PS_{5.25}Cl_{0.75}$. Its activation energy is 0.108 eV which is much smaller than that of $Li_6PS_5(BH_4)$. Thus, $Li_{6.25}PS_{5.25}(BH_4)_{0.75}$ can qualify as an ALiSIC, while $Li_6POS_4(SH)$ a near-ALiSIC. The results are summarized in Fig. 1C.

Reliability of these calculations is first reassured by noting that the calculated ionic conductivity and activation energy of $Li_{6.25}PS_{5.25}Cl_{0.75}$ are identical to those reported in the literature using the same method[8] and the diffusivities at different temperatures show good fit to the Arrhenius relation (Fig. 1C). Since the MD simulations can only include limited number of diffusion events, especially at the low temperature, we further conducted a statistical variance analysis on the data using the approach introduced in ref. [36]. As detailed in Supplementary Note 1, the relative standard deviations (RSD) found for the lowest-temperature diffusivities of $Li_{6.25}PS_{5.25}(BH_4)_{0.75}$ (@300 K) and $Li_6POS_4(SH)$ (@400 K) are 0.235 and 0.052, respectively, well below the convergence criteria of 0.3[36,37].

$Li_6POS_4(SH)/Li_{6.25}PS_{5.25}(BH_4)_{0.75}$ has a typical argyrodite structure in which the Li atoms form sulfur-centered blocks and the anion of $SH^-/BH_4^-$ anions are surrounded by these sulfur-blocks. Specifically, as shown in Fig. 2A, the sulfur end of each $SH^-$ in $Li_6POS_4(SH)$ is situated at the center of a rectangular plane defined by four lithium atoms belonging to three adjacent sulfur-blocks. The averaged S-Li bond length from the sulfur end of each $SH^-$ to the four Li-ions at the rectangular corners is $\bar{r}_{Li-SH} = 2.47(5)$. All the SH clusters are orientationally ordered in the ground state structure. Given the slight sulfur excess and $BH_4^-$ deficiency in the non-stoichiometric $Li_{6.25}PS_{5.25}(BH_4)_{0.75}$, every 2 out of 8 $BH_4^-$ sites are occupied by two doped $S^{2-}$. This is because the ionic radius of $S^{2-}$ (1.70 Å) is very similar to that of $BH_4^-$ which is between those of $Br^-$ (1.82 Å) and $I^-$ (2.06 Å)[38].

As shown in Fig. 2B, each $BH_4^-$ or doped sulfur (Sd) in $Li_{6.25}PS_{5.25}(BH_4)_{0.75}$ is tetrahedrally coordinated by four sulfur-blocks. The averaged Li-H distance between each $BH_4$ cluster and its coordinated Li-ions is $\bar{r}_{Li-H} = 2.10(10)$. The averaged Li-S distance between the doped sulfur and its coordinated Li-ions is $\bar{r}_{Li-Sd} = 2.63(20)$. Compared to the stoichiometric $Li_6PS_5(BH_4)$ which has orientationally ordered $BH_4^-$ clusters in its ground state (Fig. 1B), $Li_{6.25}PS_{5.25}(BH_4)_{0.75}$ exhibits slightly orientational disorder of $BH_4^-$ as well as positional disorder of Li-ions around the cluster and the doped sulfur (Sd), as shown in Fig. 2B.

The lithium argyrodites show the same structural features, with halogens (instead of the clusters) located in-between the sulfur-blocks. With this structure, the Li-ion can readily migrate inside each sulfur-block with an energy barrier in the order of 30 meV[11] due to the weak Li-S interaction, while it is significantly harder to migrate across neighboring sulfur-blocks. Such 'hierarchical' type of Li-ion diffusion can be directly seen in the calculated probability distribution functions (PDF) (see Method) of Li-ions. As shown in Fig. 2C, in all cases of $Li_{6.25}PS_{5.25}Cl_{0.75}$, $Li_6POS_4(SH)$ and $Li_{6.25}PS_{5.25}(BH_4)_{0.75}$, there is a relatively high probability (with greater isovalue of $P_0$) for the Li-ions to move inside each sulfur-block, while the diffusions across the sulfur-blocks mediated by the anion clusters appear with relatively low probability (with smaller isovalues). The higher probability ($0.4 \times P_0$) for the long-range Li-ion diffusion in $Li_6POS_4(SH)$ than that ($0.17 \times P_0$) of $Li_{6.25}PS_{5.25}Cl_{0.75}$ and $Li_{6.25}PS_{5.25}(BH_4)_{0.75}$ suggests that $Li_6POS_4(SH)$ has higher diffusivity at the simulation temperature (800 K) than the other two. Indeed, this is consistent with the calculated diffusivity in Fig. 1C, where the diffusivity of $Li_6POS_4(SH)$ at 800 K is about twice that of $Li_{6.25}PS_{5.25}Cl_{0.75}$ and $Li_{6.25}PS_{5.25}(BH_4)_{0.75}$, while $Li_{6.25}PS_{5.25}Cl_{0.75}$ and $Li_{6.25}PS_{5.25}(BH_4)_{0.75}$ show almost the same diffusivity at this temperature. As shown in Fig. 2C, the cross-block probability surrounding the doped sulfur in $Li_{6.25}PS_{5.25}(BH_4)_{0.75}$ (as indicated by 'S') is significantly greater than those surrounding the $BH_4$ clusters, suggesting more Li-ion diffusions near the doped sulfur in the structure. It is also found that collective motions of Li-ions in

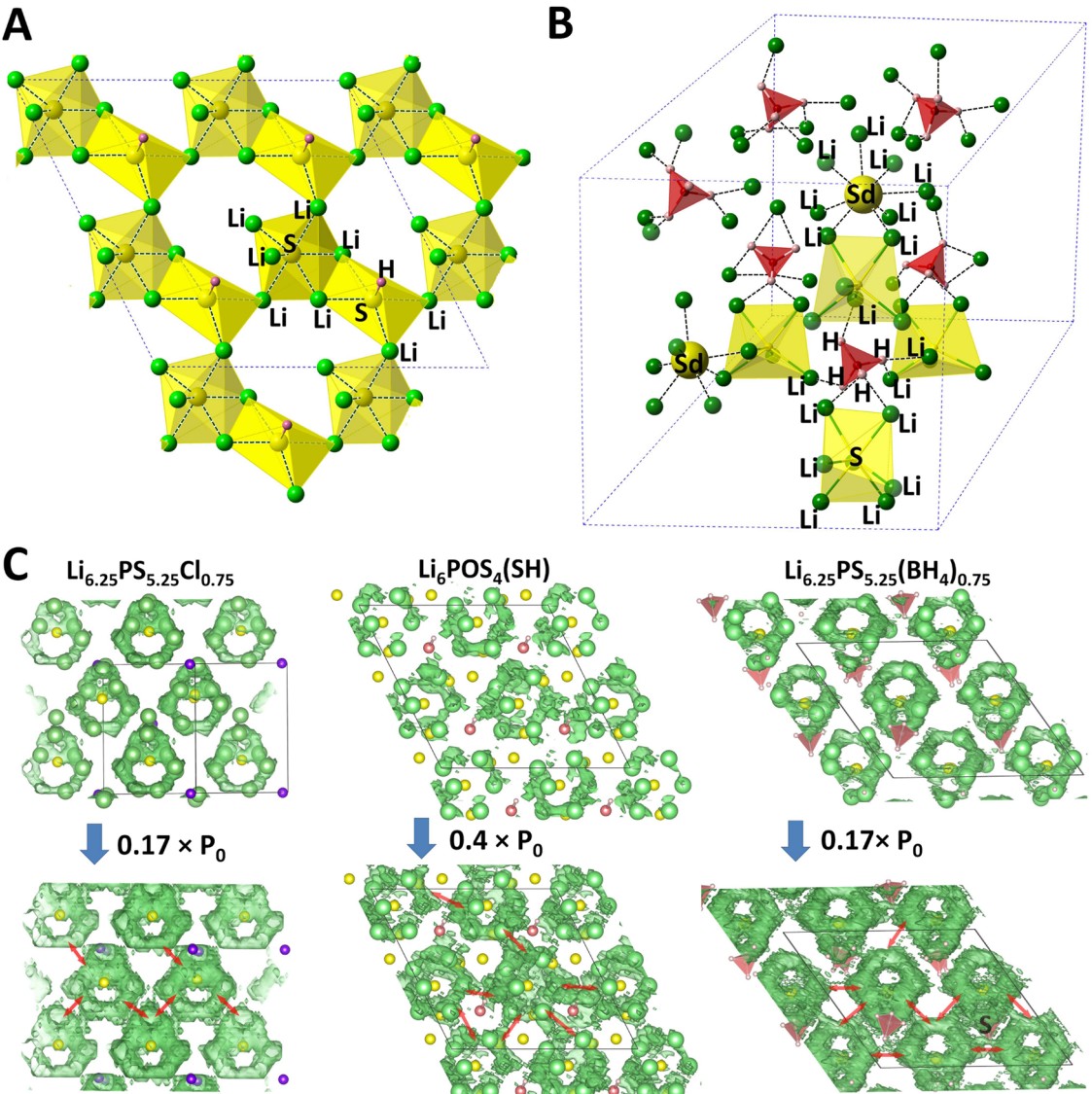

**Fig. 2 Argyrodite-type structures of the studied materials. A** Highlighted structure features of the sulfur(-centered)-blocks (in yellow) surrounding the clusters SH in $Li_6POS_4(SH)$. **B** Highlighted structure features in $Li_{6.25}PS_{5.25}(BH_4)_{0.75}$. The two doped S (Sd) in the $BH_4$ sites are put in bigger spheres. P and O atoms are omitted for clarity. **C** Calculated probability distribution function of Li-ions in $Li_{6.25}PS_{5.25}Cl_{0.75}$, $Li_6POS_4(SH)$ and $Li_{6.25}PS_{5.25}(BH_4)_{0.75}$ using the molecular dynamics simulation data at 800 K. The ground-state structures of these materials are superimposed on the probability distribution. Li atoms are in green, S in yellow and Cl in purple. Sulfur in the $SH^-$ units and the $BH_4^-$ units are highlighted in dark red, with H in pink. The upper panels with an isosurface (in green) of $P_0 = 1.8 \times 10^{-5}/Bohr^3$ show the intra-diffusions within each sulfur-block. The lower panels with relatively low probabilities show the inter-diffusions across the sulfur-blocks, as highlighted by the double-headed arrows.

$Li_6POS_4(SH)$ and $Li_{6.25}PS_{5.25}(BH_4)_{0.75}$ are found to be relatively weak. We calculated the charge-center diffusion coefficient and the Haven ratios of the materials (see Method). The inverse of Haven ratios that measure the average number of Li-ions moving collectively are 1.3 and 1.5, respectively. The calculated van Hove correlation function (see Method) of the two materials shows significantly fewer collective modes than LGPS, LLZO and LATP (Supplementary Fig. 4).

Upon thermal excitation, great rotational dynamics of $SH^-$ and $BH_4^-$ are found in $Li_6POS_4(SH)$ and $Li_{6.25}PS_{5.25}(BH_4)_{0.75}$, respectively, as shown by the extremely rapid decay of their angular time-correlation functions in Supplementary Fig. 5. The decorrelation rates of these two clusters are much higher than those of the polyanion $PS_4^{3-}$ and $B_{12}H_{12}^{2-}$ in the $Li_3PS_4$ glass[21] and $Li_2B_{12}H_{12}$[23], respectively (Supplementary Fig. 5). Therefore, the orientationally disordered phase can be readily achieved in $Li_6POS_4(SH)$ and $Li_{6.25}PS_{5.25}(BH_4)_{0.75}$ with small rotational

barriers of $SH^-$ and $BH_4^-$, estimated to be about 30 and 10 meV/ atom (see Supplementary Note 2 with Supplementary Videos 1–3), respectively. This is due to the small charge states of the mono-anions on their vertices, leading to weak interactions with their local environment. According to the Bader analysis, charges on S and H in $SH^-$ are $-0.9$ and $-0.01$e, respectively and each H in $BH_4^-$ carries only $-0.6$e charges. Given the large charge states of oxygen ($-1.5$e) and sulfur ($-1.0$e) in the much bigger polyanion $POS_3^{3-}/PS_4^{3-}$ clusters in the same structure, their estimated rotational barriers are 1–2 orders of magnitude higher than those of $SH^-$ and $BH_4^-$, and their dynamics are primarily characterized by thermal librations (see the angular time-correlation function in Supplementary Fig. 5).

To further understand the diffusion mechanisms of these advanced lithium conductors, we first study the Li-ion transport using a method previously employed to characterize non-trivial

diffusion events in LLZO[39] as well as in glassy $Li_3PS_4$[21]. Long-lived Li-ion excitation events with significant displacements can be identified by applying the functional

$$h_i(t; tp, ta, a) = \prod_{t'=ta/2-tp/2}^{ta/2} \theta(|\mathbf{r}_i(t+t') - \mathbf{r}_i(t-t')| - a) \quad (1)$$

to the MD trajectory data. The product is performed for each Li-ion in the structure snapshot at time 't', throughout the simulation time. $\theta(x)$ is the Heaviside step function equal to 1 or 0 for $x \geq 0$ or $< 0$, respectively. Therefore, only the diffusions with displacements $\geq$ 'a' that occur over a time $\leq tw = ta - tp$ and stay at distinct positions for a time $\geq tp$ will be picked out. For long-lived large-displacement events in the current study, tw = 3 and 2 ps are chosen for $Li_6POS_4(SH)$ and $Li_{6.25}PS_{5.25}(BH_4)_{0.75}$, respectively, corresponding to the time gap appeared in the van Hove correlation function $G_d(\mathbf{r}=0, t)$ that measures the typical time of one ion being replaced by another (Supplementary Fig. 4). $a = 1.8$ and 2.0 Å are chosen for $Li_6POS_4(SH)$ and $Li_{6.25}PS_{5.25}(BH_4)_{0.75}$, respectively, corresponding to the minimal distance of diffusions between the neighboring sulfur-blocks in each case (Supplementary Fig. 6).

From now on, we will call the Li-ion diffusion inside a sulfur-block as 'intra-diffusion' and the diffusion across the neighboring sulfur-block as 'inter-diffusion'. The identified diffusion events at room temperature (300 K) for all the Li-ions in the simulation cell along the time line up to 60 ps are shown in Fig. 3. A total of 14 and 16 distinctive events that are separated by significant amount of time from each other are found in $Li_6POS_4(SH)$ and $Li_{6.25}PS_{5.25}(BH_4)_{0.75}$, respectively. Many more events are found in each case with a shorter threshold distance (as shown in Fig. 3) to cover most of the intra-diffusions. Although a narrower time window and a longer threshold distance are applied for $Li_{6.25}PS_{5.25}(BH_4)_{0.75}$ than those of $Li_6POS_4(SH)$, there are significantly more events found in $Li_{6.25}PS_{5.25}(BH_4)_{0.75}$, which is consistent with its relatively high ionic conductivity. Similarly, as shown in Supplementary Fig. 7, the identified number of long-lived large-displacement diffusion events in the stoichiometric $Li_6SP_5(BH_4)$ is significantly smaller than that of the non-stoichiometric $Li_{6.25}SP_{5.25}(BH_4)_{0.75}$, which explains its much lower ionic conductivity than that of the latter.

To obtain a complete physical picture of the Li-ion transport in each material, we conducted a careful analysis and combined all the related individual events together in time and space. For $Li_6POS_4(SH)$, its individual events shown in Fig. 3A turn out to be only parts of three combined events (Event 1, 2 and 3) as shown in Fig. 4. For $Li_{6.25}SP_{5.25}(BH_4)_{0.75}$, its individual events shown in Fig. 3B turn out to be only parts of four combined events (Event 4, 5, 6 and 7) as shown in Fig. 5. We put the detailed description for each event as well as the description for the correlation between the Li-ion displacements and cluster dynamics in Supplementary Note 3.

By going over these combined events of both materials, we can elucidate the paddle-wheel effect in the context of the current materials and establish some new diffusion mechanisms. First, by Li–Li repulsion, a transporting Li-ion can relay its kinetic energy from one to another, leading to a cascade of diffusion events and long-distance conduction. We call this the 'billiard-ball' mechanism. It has been long believed that the local intra-diffusions (e.g., within a sulfur-block of the argyrodite structure) will not contribute to the long-distance conductivity. However, through the billiard-ball mechanism, the following behaviors are observed (please refer to Figs. 4–5 as well as the descriptions in Supplementary Note 3): An intra-diffusion can trigger a large-displacement inter-diffusion across the sulfur-blocks (e.g., Li(27)-Li(17) in Event 3 and Li(24)-Li(28) in Event 7);

An inter-diffusion, either triggered by an intra-diffusion or by direct thermal excitation (e.g., Li(48) in Event 2, Li(13) in Event 3 and Li(4) in Event 6), can trigger a set of intra-diffusions in other sulfur-blocks (e.g., Li(27)-Li(17)-Li(5) in Event 3, Li(24)-Li(28)-Li(27,15) in Event 7, Li(48)-Li(32) in Event 2, Li(13)-Li(39) in Event 3 and Li(4)-Li(16) in Event 6); An intra-diffusion triggered by an inter-diffusion event can trigger intra-diffusion of another Li-ion in the same sulfur-block, e.g., Li(27)-Li(17)-Li(5)-Li(46) in Event 3 and Li(4)-Li(16)-Li(46) in Event 6.

Second, there is a relay of intra-diffusions in the same sulfur-block due to the Li-Li repulsion, which does not lead to any inter-diffusion and cross-block conduction, as shown by Li(49)-Li(32)-Li(36)-Li(14) in Event 5 (Fig. 5 and Supplementary Note 3). We call this the 'revolving-door' mechanism.

Third, another interesting mechanism is the 'docking-undocking' of a Li-ion. After a large-displacement diffusion out of its sulfur block, a Li-ion in $Li_6POS_4(SH)$ can be trapped by interacting strongly with a SH cluster and stay away from any diffusion event for a great amount of time (over tens of pico-seconds), as shown by Li(48) in Event 2 (Fig. 4 and Supplementary Fig. 8). Clearly, this will be an adverse factor for the lithium conduction of the material. On the other hand, a Li-ion originally interacting strongly with a SH cluster and belonging to no sulfur-blocks can later be released to participate in diffusions in a nearby sulfur-block and eventually reside in that block, as shown by Li(19) in Event 2 (Fig. 4, Supplementary Note 3 and Supplementary Fig. 9). No long-time 'docking' event of Li-ion to a $BH_4$ cluster or a doped sulfur is found in the analysis of $Li_{6.25}PS_{5.25}(BH_4)_{0.75}$, which is an advantage over the case of $Li_6POS_4(SH)$. There are some cases in which a vacated site due to an intra-diffusion is later occupied by another intra-diffusion Li-ion, as shown by Li(41)-Li(2) in Event 1, Li(10)-Li(32) in Event 2 (Fig. 4 and Supplementary Note 3) and Li(43)-Li(39) in Event 4 (Fig. 5 and Supplementary Note 3). Such behavior is also known in other types of ionic conductors.

Fourth, the paddle-wheel effect found in $Li_6POS_4(SH)$ and, especially in $Li_{6.25}PS_{5.25}(BH_4)_{0.75}$, are not quite the same as that in the lithium conductors with heavy polyanions (e.g., $Li_3PS_4$ in ref. [21]). One consequence of the paddle-wheel effect is that the rotation of a cluster can 'drag/pull' its nearby Li-ion to move along. This suggests very strong interaction between the cluster and the Li-ion, which is not likely to be favored by a mono-anion with small charge states on its vertices (e.g., $BH_4^-$) as compared to polyanions (e.g., $PS_4^{3-}$). It also suggests that the motion of the Li-ion will not significantly influence the dynamics of the cluster, which is not favored by a light and small cluster (e.g., $SH^-$ or $BH_4^-$) as compared to a heavy and large cluster such as $PS_4^{3-}$ or $B_{12}H_{12}^{2-}$. This suggests that a pronounced rotation of the cluster should lead or precede the related motion of the Li-ion. However, it would be difficult to have a clear picture of this in a complex system, when pronounced rotational dynamics can be readily thermally excited throughout the time (as in the current cases of $SH^-$ and $BH_4^-$) and the Li-ion is interacting with multiple clusters simultaneously. In the studied materials, it is hard to find a clear case in which the Li-ion displacement is initiated by a rotation of the cluster, as shown by the SH and $BH_4$ dynamics in Figs. 4 and 5. Rather, pronounced rotational dynamics are found after or, in some cases, during the displacement of the Li-ion. This suggests that the rotational dynamics of the clusters are more of a response to the motion of the Li-ion. In addition, there are strong thermally excited (rotational and translational) cluster dynamics in these materials, especially in $Li_{6.25}PS_{5.25}(BH_4)_{0.75}$, which significantly fazes the correlation, as shown by the big fluctuations in the rotational dynamics of SH and $BH_4$ clusters in Figs. 4 and 5, respectively. It is also found that the correlation between the Li-ion displacement and the cluster dynamics is more pronounced in $Li_6POS_4(SH)$ than in $Li_{6.25}PS_{5.25}(BH_4)_{0.75}$, as

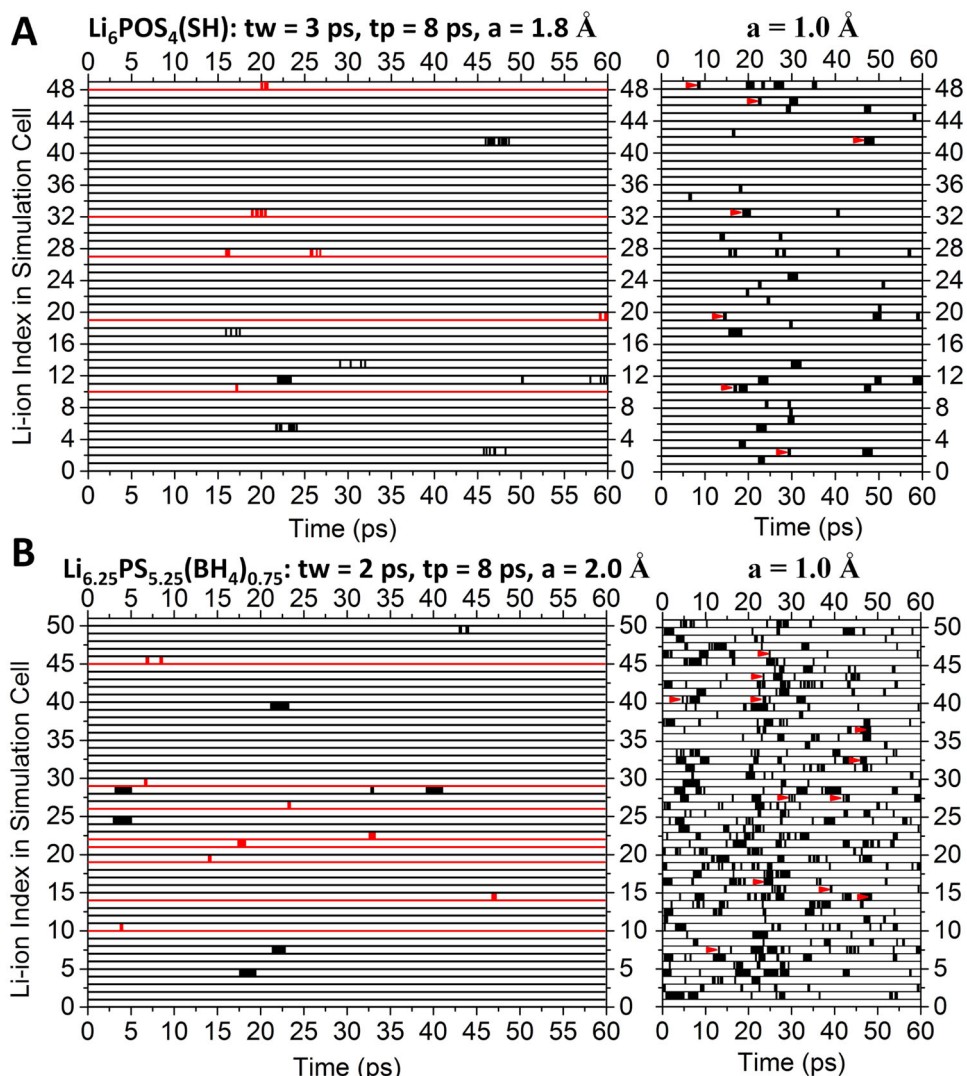

**Fig. 3 Identified Li-ion diffusion events at room temperature (300 K) using Eq. (1). A** The diffusion events identified in the simulation cell of $Li_6POS_4(SH)$ under the stated condition, with a total of 48 Li-ions (Left panel). For example, for the 48th Li-ion (corresponding to the line at '48'), there is such a recorded event starting at about 20 ps. Those events with a narrow time window (tw) of 2 ps are marked in black. With the threshold distance (a) lowered to 1.0 Å, many more events are picked out (Right panel). The events that are related to a major diffusion event belonging to a long-time line are marked by red arrows and will be included later in Fig. 4. **B** The diffusion events identified in the simulation cell of $Li_{6.25}PS_{5.25}(BH_4)_{0.75}$ under the stated condition, with a total of 50 Li-ions (Left panel). Those events with a narrow time window (tw) of 1 ps are marked in black. With the threshold distance (a) lowered to 1.0 Å, many more events are picked out (Right panel). The events that are related to a major diffusion event belonging to a long time-line are marked by red arrows and will be included later in Fig. 5.

shown by the rotational profile of the cluster in resemblance to that of the Li-ion displacement (e.g., Li(48)-SH(8) in Fig. 4D, Li(11)-SH(4,8) in Fig. 4E, Li(19)-SH(3) in Fig. 4F, Li(27)-SH(3) in Fig. 4G, Li(17)-SH(3) in Fig. 4H and Li(13)-SH(7) in Fig. 4I; also see Supplementary Note 3). This is due to the fact that $SH^-$ has a higher (Bader) charge state of its S ($-0.9e$), a much heavier weight and significantly less rotational freedom under thermal excitation (judged by its angular decorrelation rate in Supplementary Fig. 5) than those of $BH_4^-$.

Thus, besides the commonly known paddle-wheel effect that only emphasizes on the ionic diffusion promoted by the cluster rotation, a more complete picture emphasizing on the interplay between the ion transport and the cluster dynamics (both rotational and translational) is formed. Here, the cluster dynamics can be categorized as 'active' (rotational and translational) dynamics due to the thermal excitation of the cluster, and 'responsive' dynamics due to the cluster's reaction to a passing Li-ion. Given

the great freedom of SH and $BH_4$ clusters in their corresponding systems, their (thermally-excited) 'active' dynamics and their 'responsive' dynamics to the nearby transporting Li-ion are always entangled. One needs to separate the impacts of these two kinds of dynamics for a better understanding. To achieve this, we device a set of calculations on the distinctive Li-ion migration routes in each structure (Fig. 6), according to the routes of both intra- and inter-diffusions observed in the studied Events.

For the effect of the 'responsive' dynamics, we conduct two sets of calculations for the low-energy pathway along each route using the nudged elastic band (NEB) method. One is to fix all the atoms except the migrating Li-ion. The other is to only allow the Li-ion and its interacting cluster (or doped-sulfur for some cases in $Li_{6.25}PS_{5.25}(BH_4)_{0.75}$) to move, so that the cluster can relax and respond to the Li-ion movement during its migration. The difference between these two sets of calculations can show how the responsive dynamics of a cluster can facilitate (or hinder) the

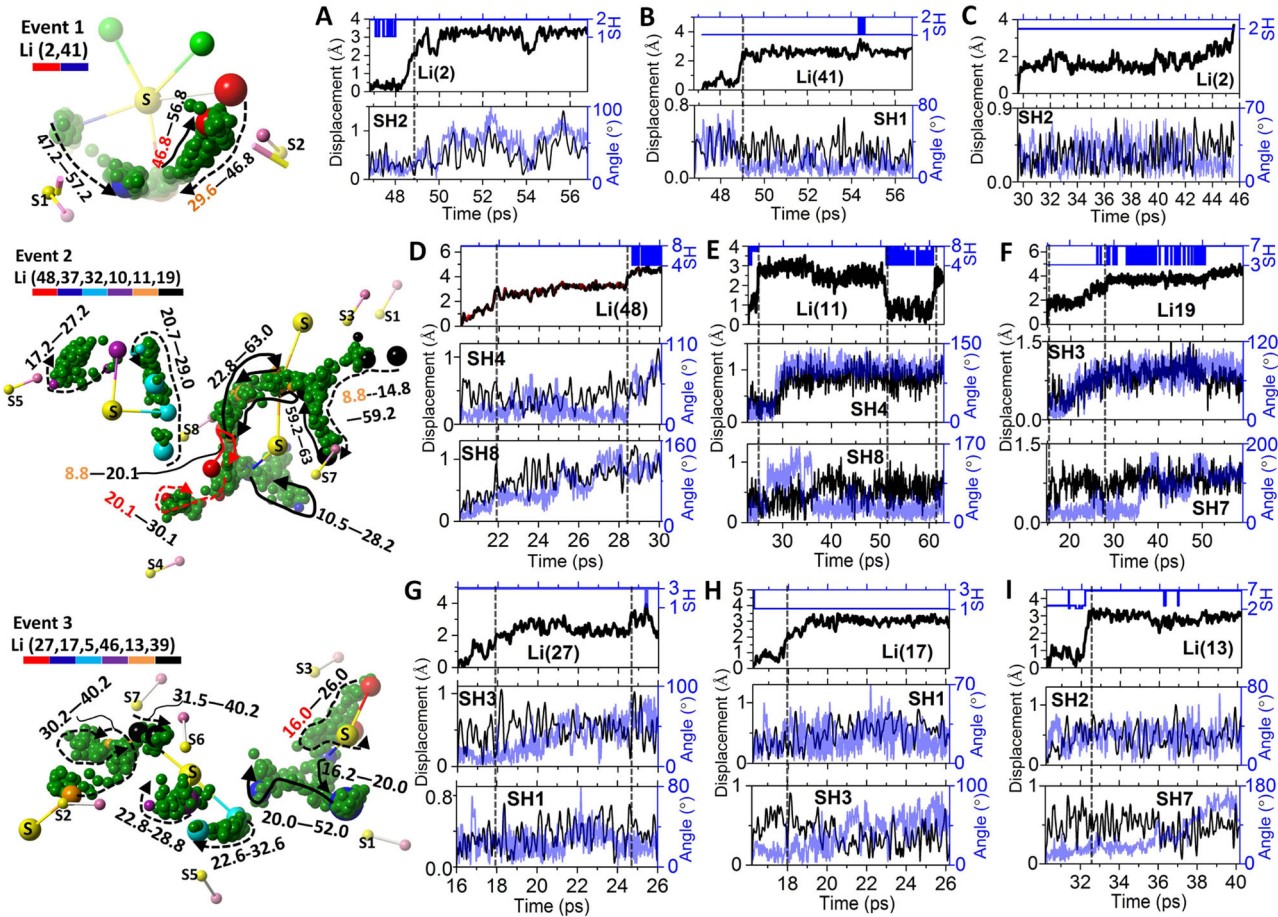

**Fig. 4 The three established major events of Li$_6$POS$_4$(SH) after analyzing the individual events found in Fig. 3A.** In each Event (1, 2, and 3), the involved Li-ions (in green), the S (in yellow) centers of the sulfur-blocks, the related SH (H in pink) clusters and the trajectories (in dark green) of the Li-ions are included. For the purpose of clarity, irrelevant atoms are omitted in the Event. The Li-ions participating in the diffusions and some key pointers along their trajectories are highlighted by different colors according to the legend. The arrow lines show the route vectors of the trajectories. The numbers corresponding to the 'beginning–ending' time of the trajectory data. The 'current time' of the frame in each event is highlighted in red. A 'history time' is present in orange when a historical event is to be discussed. **A** Calculated displacements of the 2nd Li-ion, Li(2), and the 41st Li-ion, Li(41), in the time periods of 46.8–56.8 ps and 47.2–57.2 ps, respectively, corresponding to the trajectories in Event 1. The index of the nearest SH cluster to the Li-ion at every time point along the time-line is identified. For example, both SH(1) and SH(2) are found to interact with Li(2) at some time points during its diffusion from 46.8 to 56.8 ps. **B** Calculated angular (Angle in degrees) and the translational displacement of the SH cluster from 46.8 to 56.8 ps, during the diffusions of Li(2) and Li(41). **C** Calculated displacement of the 'historical' diffusion event of Li(2) from 29.6 to 46.8 ps, which leads to the current event. The angular and translational displacements of its related SH(2) cluster during this time period are also shown. **D–F** Calculated displacements of Li(48), Li(11) and Li(19) in Event 2 and the dynamics of their related SH clusters. **G–I** Calculated displacements of Li(27), Li(17) and Li(13) in Event 3 and the dynamics of their related SH clusters.

migration of the Li-ion. Meanwhile, the calculation can also show which migration route is energetically preferable.

To study the effect of the 'active' dynamics of cluster, we manually rotate the cluster to different angles about its high-symmetry axes. Only the cluster under study is rotated, while the other ones in the supercell are fixed to their ground-state orientation(s). With each rotated angle of the cluster, we calculate the low-energy pathway for each migration route. All atoms except the migrating Li-ion are fixed during the calculation. This is to simulate the 'active' rotation of a cluster due to thermal excitation and see how the rotational angle can change the energy pathway of the Li-ion. Specifically, how such an 'active' rotation can change the relative energy between the starting and ending sites of the migration, as well as the energy extreme of the transition state along the pathway. Note that, due to the selective dynamics and the fixed volume of the cell during these NEB calculations, the obtained energy barriers are expected to be significantly overestimated. However, it is adequate for our purpose here to draw qualitative conclusions.

Figure 6A shows the studied migration routes in the Li$_6$POS$_4$(SH) structure, where P1, P2, and P3 are three distinctive intra-diffusion routes while P4 and P5 are two distinctive inter-diffusion routes. The results for the effects of responsive dynamics of cluster are shown in Fig. 6B. These include the calculated energy barriers with fixed atoms except the migrating Li-ion (denoted with 'F') and the energy barriers allowing the interacting cluster (or the interacting doped sulfur) to relax (denoted with 'M'). It is found that, in all cases, allowing the cluster (or doped sulfur) to relax and respond to the moving Li-ion will greatly lower (up to 80%) the migration barrier, for both intra- and inter-diffusions, i.e., P1(F) vs. P1(M), P2(F) vs. P2(M), P4(F) vs. P4(M) and P5(F) vs. P5(M). Note that the intra-diffusion route P3 is away from any cluster. Therefore, its energy barrier is barely related to the cluster dynamics. The inset of Fig. 6B shows the trajectory of the typical responsive dynamics of an SH cluster to a passing Li-ion (in green). The above results show the cluster responding to the

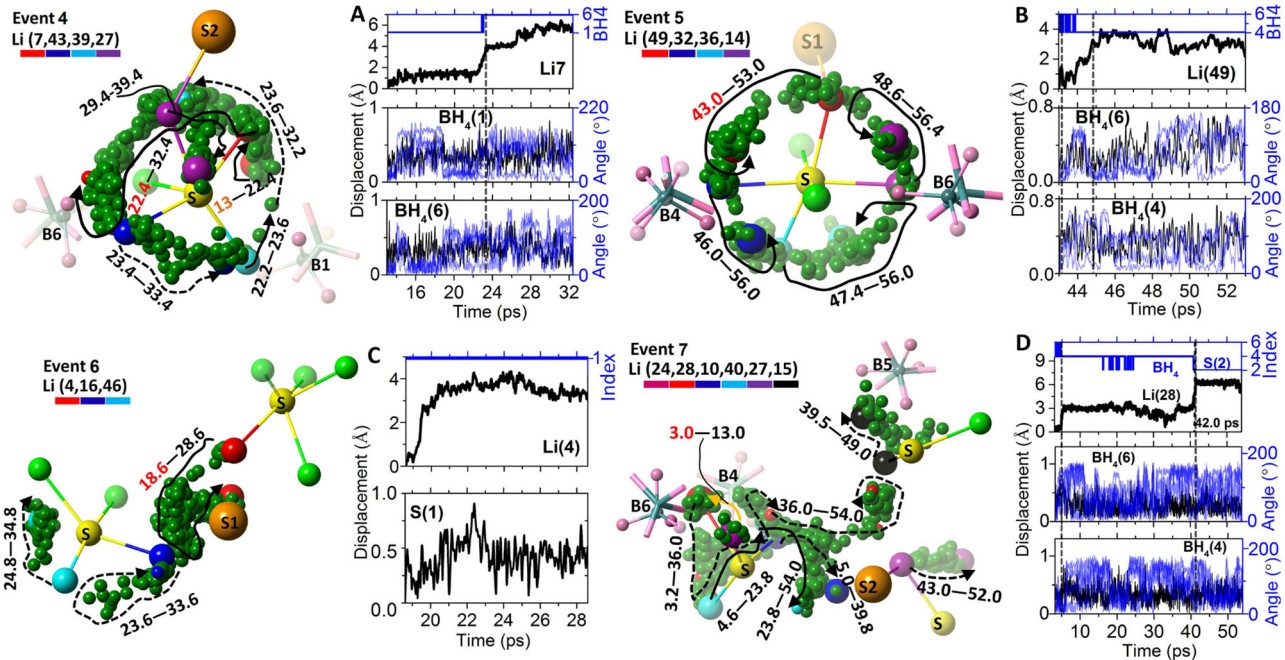

**Fig. 5 The four established major events of Li$_{6.25}$PS$_{5.25}$(BH$_4$)$_{0.75}$ after analyzing the individual events found in Fig. 3B.** In each Event (4, 5, 6, and 7), the involved Li-ions (in green), the S (in yellow) centers of the sulfur-blocks, the doped sulfur (in orange), the related BH$_4$ (B in dark blue and H in pink) clusters and the trajectories (in dark green) of the Li-ions are included. For clarity, irrelevant atoms are omitted in the Event. The Li-ions participating in the diffusions and some key pointers along their trajectories are highlighted by different colors according to the legend. The arrow lines show the route vectors of the trajectories. The numbers corresponding to the 'beginning--ending' time of the trajectory data. The 'current time' of the frame in each event is highlighted in red. A 'history time' is present in orange when a historical event is to be discussed. **A** Calculated displacements of Li(7) corresponding to the trajectory in Event 4. The Inset shows the index of the nearest BH$_4$ cluster of the Li-ion at a particular time. The rotational (angular changes of the four B-H unit vectors, in blue lines) and the translational (in black lines) displacements of the related BH$_4$ clusters are also given. **B** Calculated displacements of Li(49) corresponding to the trajectory in Event 5. Also given are the rotational and translational displacements of the related BH$_4$ clusters. **C** Calculated displacement of Li(4) corresponding to the trajectory in Event 6. This Li-ion interacts with a doped sulfur (S1) instead of BH$_4$ clusters. Therefore, the translational displacement of the doped sulfur is given. **D** Calculated displacement of Li(28) corresponding to the trajectory in Event 7. The rotational and translational displacements of its related BH$_4$ clusters are given.

passing Li-ion via its rotational and translational dynamics which will always lower the energy barrier.

It is also found that the intra-diffusion routes of P1 and P2 show the lowest migration barriers (in the order of 10 meV) of all. P1 is characterized by the Li-ion interacting with one SH cluster (SH1) along its pathway, and P2 characterized by the Li-ion migrating between two SH clusters (SH1 and SH2). Their low migration barriers explain why such intra-diffusions can be readily thermally-excited and why all the intra-diffusions in the studied Events belong to the P1 and P2 types. For example, as shown in Fig. 4, Li(2)-SH(2) and Li(41)-SH(1) in Event 1 belong to P1; Li(19) migrating between SH(3) and SH(7) in Event 2 belongs to P2; and Li(27)-SH(3) belongs to Event 3. As expected, the calculated migration barriers of the inter-diffusion routes P4 and P5 are significantly greater than those of P1 and P2. This explains why only a few inter-diffusions are observed in the studied Events, with only Li(48) in Event 2 and Li(27) in Event 3 in Fig. 4. Both of these inter-diffusions are triggered by intra-diffusions. The above findings suggest that the 'billiard-ball' mechanism, featured by realizing long-range ionic conduction by the relay of local ionic diffusions, is the origin of the high ionic conductivity at low temperatures.

The results for the effects of active dynamics of cluster is shown in Fig. 6C. The SH cluster is rotated up to 300 degrees with the trajectory of its sulfur end shown in the inset of Fig. 6C. For each route, the relative energy difference (Diff) between the starting and the ending sites of the migration (in dashed line) as well as the energy extreme (Ext) along the route (in solid line) are calculated.

It is found that the active rotation of a cluster can significantly change the pathway energy of Li-ion, with greatly lowered barrier at some angles and greatly increased barrier by the other, as shown by P1(Ext), P2(Ext), P4(Ext) and P5(Ext) in Fig. 6C. For example, when SH rotates to 125 degrees, the energy barrier of P1(Ext) is lowered by about 70%, while increased by about 80% when the cluster is further rotated to 230 degrees. The relative energy between the starting and ending sites of a route is also changed by the active rotation, as shown by P1(Diff), P2(Diff), P4(Diff) and P5(Diff) in Fig. 6C. These suggest that an active rotation of a cluster to certain angle can either trigger/promote or inhibit/suppress the ionic diffusion along certain route. Therefore, compared to the responsive dynamics of cluster which will always facilitate the Li-ion diffusion, the thermally excited active rotation will introduce some randomness into the ionic-transport picture. Figure 6C also shows that the effect of the active rotation is route-dependent. For example, the active rotation can change the pathway energies of P1(Ext) and P4(Ext) much more dramatically than it can change those of P2(Ext) and P5(Ext).

Similar results are found for the responsive and active dynamics of BH$_4$ in Li$_{6.25}$PS$_{5.25}$(BH$_4$)$_{0.75}$ for the migration routes shown in Fig. 6D. The effect of the responsive dynamics of the BH$_4$ cluster is more pronounced than that of SH, as shown in Fig. 6E. Note that, for the P1 route, if we fix the boron atom of the cluster and only allow the hydrogen atoms to move, the lowered energy of P1 becomes less, as shown by the dotted line vs. the solid line in red in Fig. 6E. This suggests that the translational dynamics of the cluster also contribute to the responsive

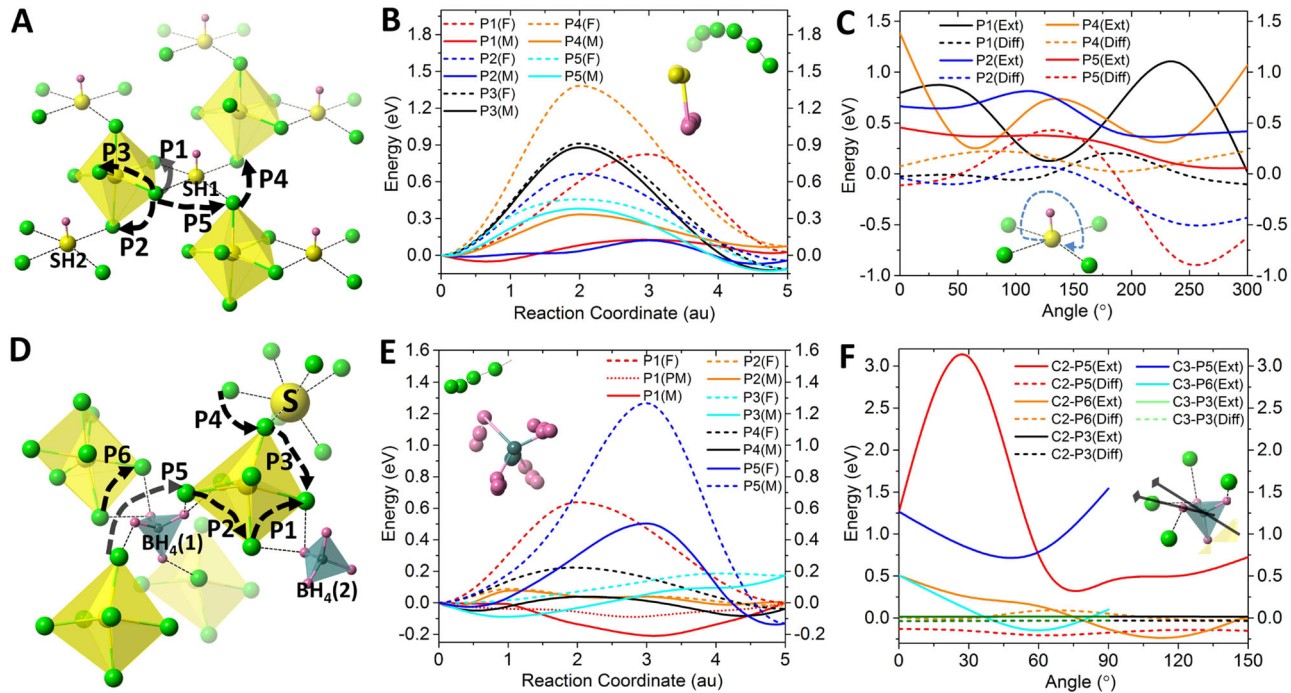

**Fig. 6 Studies to understand the respective effects of the 'responsive' dynamics and the 'active' rotation of the clusters. A** The distinctive migration routes in the structure of $Li_6POS_4(SH)$. **B** The calculated migration barriers for different routes with (F) and without (M) the fixed dynamics of the cluster. The inset shows the rotational dynamics of a SH cluster responding to the movement of a Li-ion to lower the energy along the pathway. **C** Calculated energy difference (Diff) between the starting and the ending sites of a migration route as well as the energy extreme (Ext) of the transition state in the pathway vs. different rotational angles of the SH cluster. The inset shows the rotation trajectory of the sulfur end of the cluster. **D** The distinctive migration routes in the structure of $Li_{6.25}PS_{5.25}(BH_4)_{0.75}$. **E** The calculated migration barriers for different routes with and without the fixed dynamics of the cluster. The inset shows the rotational dynamics of a $BH_4$ cluster responding to the motion of a Li-ion to lower the energy along the pathway. **F** Calculated energy difference (Diff) between the starting and the ending sites of a migration route as well as the energy extreme (Ext) of the transition state in the pathway vs. different rotational angles of the $BH_4$ cluster about its $C_2$ and $C_3$ axes as shown in the inset.

dynamics of cluster significantly. The P2 route between the two $BH_4$ (1) and (2) clusters, and the P3 and P4 routes that are around the doped sulfur exhibit the flattest migration barriers. This explains why the P2-type intra-diffusions between two clusters are widely found in the studied Events, such as those between the B1 and B6 clusters in Event 4 as well as between the B4 and B6 clusters in Events 5 and 7 in Fig. 5.

In addition to their flatness, the energy barriers of both P3 and P4 will not be affected by any active rotations of the $BH_4$ (as demonstrated by P3 in Fig. 6F), since both P3 and P4 are unrelated to any $BH_4$ cluster as shown in Fig. 6D. This explains why all the major events identified for $Li_{6.25}PS_{5.25}(BH_4)_{0.75}$ involve the doped sulfur, with Event 6 involving only the doped sulfur and no $BH_4$ cluster at all, as shown in Fig. 5. Note that there are only two doped sulfur sites out of eight $BH_4$ sites in the simulation cell. This suggests that the doped sulfur sites play a critical role for the ionic diffusion in $Li_{6.25}PS_{5.25}(BH_4)_{0.75}$. Indeed, the stoichiometric $Li_6PS_5(BH_4)$ exhibits much lower ionic conductivity than those of $Li_{6.25}PS_{5.25}(BH_4)_{0.75}$ and $Li_6POS_4(SH)$ (Fig. 1C). This may be explained by its lack of the doped sulfur sites, leading to fewer Li-ion diffusion events as shown in Supplementary Fig. 6 and being more susceptible to the adverse effects that could be brought by the active rotations of $BH_4$ due to excessive thermal excitation of the clusters found in $Li_6PS_5(BH_4)$ (Supplementary Fig. 10).

## Discussion

Several key conclusions can be drawn from the above. First, the origin of the high ionic conductivity of the materials at low temperatures is attributed to a new mechanism which we term

'billiard-ball' mechanism. It was believed that local ionic diffusions cannot contribute to the long-range ionic conduction of a material. However, the 'billiard-ball' mechanism shows that long-range ionic conduction can be effectively realized by a relay of local diffusions across the lattice sustained by Li-Li repulsions. Such mechanism is also found for the ionic conduction of the nonstoichiometric lithium argyrodite $Li_{6.25}PS_{5.25}Cl_{0.75}$, as shown in Supplementary Fig. 11. To sustain the 'billiard-ball' mechanism, it is important for the Li-Li repulsion to overcome the pathway barrier. This can be supported by four factors in the studied systems: (1) the low energy barriers of intra-diffusions in a sulfur block; (2) the responsive (rotational and translational) dynamics of cluster to lower the energy barriers for local diffusions as demonstrated in $Li_{6.25}PS_{5.25}(BH_4)_{0.75}$ and $Li_6POS_4(SH)$; (3) doped sulfur sites as weak interacting conduit to lower the energy barriers for local diffusions as demonstrated in $Li_{6.25}PS_{5.25}(BH_4)_{0.75}$ and $Li_{6.25}PS_{5.25}Cl_{0.75}$; (4) Li-ion excess to enhance the Li-Li repulsive interaction in the nonstoichiometric configurations $Li_{6.25}PS_{5.25}(BH_4)_{0.75}$ and $Li_{6.25}PS_{5.25}Cl_{0.75}$. The 'billiard-ball' mechanism is especially useful to achieve high ionic conductivity at low temperatures and should be considered in the future design of solid electrolytes. Other uncovered mechanisms in this study, including the 'revolving-door' mechanism and the 'docking-undocking' mechanism with significant time scale (over tens of pico-seconds) can be also helpful for our understanding of ionic diffusion process in solids.

Secondly, besides the paddle-wheel effect that merely emphasizes on the Li-ion migration promoted by the cluster rotation, the current study shows a more complete picture of the dynamic correlation between the cluster-ions and Li-ions with the

following new aspects: (1) The dynamics of cluster can be categorized into 'responsive' and 'active' dynamics. The former is characterized by accommodating the Li-ion movement and will always lower the migration barrier. The latter is characterized by active rotation of clusters due to thermal excitation, which, depending on the migration route, may facilitate or inhibit the ionic diffusion. (2) Both translational and rotational degrees of freedom (not just the rotation) of the cluster can contribute significantly to its dynamic correlation with the Li-ion.

Thirdly, the study shows that incorporating small mono-anion clusters into a modern fast-ion framework can serve as a powerful strategy to achieve lithium superionic conductors with high transport numbers. Mono-anion clusters, such as $OH^-$, $SH^-$, $CN^-$ and $BH_4^-$, have similar ionic radii compared to the halogen group elements. Therefore, any ionic-conducting structures that can accommodate halogen site can be potential candidates for such a strategy. Indeed, we have seen this realized experimentally in Li/Na-rich antiperovskites $(Li/Na)_{3-x}(O/S)X$ (X = halogen or cluster)[40,41] and lithium argyrodites[42,43]. More recently discovered solid electrolyte systems with good properties, such as $Li_xScCl_{3+x}$[37], $Na_{3-x}Y_{1-x}Zr_xCl_6$[44] and $Na_{3-y}PS_{4-x}Cl_x$[45], can be also subject to such a strategy. The currently reported ALiSIC $Li_6POS_4(SH)$ and $Li_{6.25}PS_{5.25}(BH_4)_{0.75}$ are likely to be synthesizable given their relatively low formation energies and the fact that both $OH^-$ and $BH_4^-$ are frequently used in the material synthesis[40,46]. In fact, by a mechanochemical process, a glassy matrix of $(100-x)\cdot(0.75Li_2S\cdot0.25P_2S_5)\cdot xLiBH_4$ containing a crystalline phase of $Li_5PS_4(BH_4)_2$ (with $BH_4$ excess) has been successfully prepared most recently, with $BH_4^-$ clusters at the halogen sites of the argyrodite[43]. Such system already exhibits a high RT ionic conductivity of about 2 mS/cm with a small activation energy of 0.166 eV[43]. These early results are consistent with the calculations in the current study and very encouraging for the ultimate synthesis of the ALiSIC.

## Methods

**First-principles structure search for the studied lithium conductors**. The cluster-based structures are searched using Particle Swarm Optimization (PSO) algorithm (based on the PSO library of CALYPSO[47]) and density functional theory. The cluster-ions are treated as a regular 'element' at the beginning of the search, initially introduced into the system with their gas-phase geometries (Supplementary Fig. 12) and other constituents as single-atomic species. For example, to search the structure of $Li_6PS_5(BH_4)$, four atomic elements, including Li, P, S and $BH_4$, are used to form the trial structures containing up to four formula units. Note that $BH_4$ here is treated as a whole, instead of independent B and H atoms, by defining its gas-phase geometry using internal coordinates (Z-matrix). In a normal search, a population of 30–40 initial crystal structures that contain certain cluster-ion are generated according to different symmetries and number of atoms in the unit cell. These structures are then subjected to full optimization (in both lattice parameters and atomic positions) until prescribed convergence criteria are met (energy difference $\leq 10^{-5}$ eV and forces $\leq 0.01$ Å/eV). A total of 60% of the optimized structures showing the lowest energies form the next generation with newly generated structures according to symmetries. After 10 to 30 generations of optimizations, structures with the lowest energies do not change and are picked out as the candidates for further studies.

**Calculations of the basic properties of the lithium conductors**. Density functional theory (DFT) calculations are carried out with Perdew–Burke–Ernzerhof (PBE) generalized gradient approximation (GGA)[48] implemented in the VASP package[49]. The projector augmented wave (PAW) pseudopotential method[50] is used. A tested dense Monkhorst-Pack **k**-point mesh ($5 \times 5 \times 5$ in a typical case with lattice parameters around 6–7 Å) is used in each calculation. The cutoff energy is 550 eV. The energy convergence is set to $10^{-6}$ eV and the force convergence is set to 0.001 eV/Å. Structures of the lithium conductors are fully optimized without any symmetry constraint. The phonon dispersion relations of the materials are calculated using the Density Functional Perturbation Theory (DFPT). Geometry of the unit cell is optimized with an energy convergence of $10^{-8}$ eV and force convergence of $10^{-4}$ eV/Å. Phonon frequencies are first calculated on a dense **q**-grid. Frequencies at other **q** points are then interpolated from the calculated ones. Electronic structures of the materials are calculated using HSE06 functional which is known to yield reliable band gap values. Bader charge analysis is used to obtain the charge states of different species in the materials.

**Diffusion and correlation analysis based on the molecular dynamics simulations of the lithium conductors**. Ab initio molecular dynamics (AIMD) simulations are conducted using $2 \times 2 \times 2$ supercells of the studied lithium conductors with a time step of 2 fs. Typical AIMD simulations last over 100 ps until the extracted Li-ion diffusion coefficients are converged and show good linearity when fitted to the Arrhenius relation (see below). The first 10–20 ps are allowed for the system to reach thermal equilibrium before the collection of the structural data at the following time steps. NVT ensemble (with constant volume) is used at different simulated temperatures to speed up the ion hopping process. The diffusivity (D) at each temperature is obtained by linear fitting to the calculated mean squared displacement (MSD) from the collected MD data according to

$$D = \lim_{t \to \infty} \frac{1}{2dt} \frac{1}{N} \sum_{i=1}^{N} \left\langle \left[ \boldsymbol{r}_i(t) - \boldsymbol{r}_i(0) \right]^2 \right\rangle,$$

where $\left[ \boldsymbol{r}_i(t) - \boldsymbol{r}_i(0) \right]$ is the displacement vector of $Li^+$ at time $t$. The ionic conductivity is then calculated from the Nernst-Einstein relation

$$\sigma = D(Ne^2/k_BT)$$

with $N$ the number of ions per $cm^3$. Other symbols have their customary meanings. The Arrhenius relationship

$$D = A\exp\left(\frac{E_a}{k_BT}\right)$$

is used to fit to the diffusivities at different temperatures and extrapolate to the value at room temperature. The pre-factor $A$ and activation energy $E_a$ are the fitting parameters in the relation. The diffusivity that measures the diffusion of the charge center of all the Li-ions (rather than individual Li-ion) in the system is defined as

$$D_c = \lim_{t \to \infty} \frac{1}{2dt} \left\langle \left[ \frac{1}{N} \sum_{i=1}^{N} \boldsymbol{r}_i(t) - \frac{1}{N} \sum_{i=1}^{N} \boldsymbol{r}_i(0) \right]^2 \right\rangle.$$

The Haven ratio is defined as $D/(ND_c)$ and its inverse is defined as the correlation factor.

The distinct van Hove time correlaton function is defined as

$$G_d(\boldsymbol{r}, t) = \frac{1}{N} \left\langle \sum_{i \neq j}^{N} \delta\left[ \boldsymbol{r} + \boldsymbol{r}_j(0) - \boldsymbol{r}_i(t) \right] \right\rangle$$

which measures the probability of finding, at time $t$, another ion at distance $\boldsymbol{r}_i$ given the first ion at $\boldsymbol{r}_j$ at $t = 0$. $G_d(\boldsymbol{r}, t = 0)$ is the pair distribution function of the structure. $G_d(\boldsymbol{r} = \boldsymbol{0}, t)$ measures the probability of one ion being replaced by another in time $t$.

The probability distribution functions for different species in the material are computed using the MD data at 800 K. Our code can read through the snapshots of the material structure over a specified time period and count the probability for a species present at each grid point in the 3D space.

## Data availability

All data are available in the paper and from the author upon request.

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

## Acknowledgements

The work is supported by the U.S. Department of Energy, Energy Efficiency and Renewable Energy award number DE-EE0008865 (P.J. and H.F.) and the U.S. Department of Energy, Office of Basic Energy Sciences, Division of Materials Sciences and Engineering under Award DE-FG02-96ER45579 (P.J.). Resources of the National Energy Research Scientific Computing Center supported by the Office of Science of the U.S. Department of Energy under Contract No. DE-AC02-05CH11231 is also acknowledged (P.J. and H.F.).

## Author contributions

H.F and P.J. designed the project. H.F. conducted the research. H.F. and P.J. wrote the paper.

## Competing interests

The authors declare no competing interests.
