## [Peer Review File · Nature Communications]

REVIEWER COMMENTS

Reviewer #1 (Remarks to the Author):

This manuscript presents the computational study on a universal theory of the principle and mechanism of superionic lithium-ion conduction at atomic scale and suggests new argyrodite-type solid electrolytes which have very high ionic conductivity. Through ab initio molecular dynamics (AIMD) simulations, they show that $\text{Li}_6.25\text{PS}_5(\text{BH}_4)_{0.75}$ and $\text{Li}_6\text{POS}_4(\text{SH})$ have super-high ionic conductivities of 177 and 82 mS/cm with low activation energies of 0.108eV and 0.166eV, respectively, at room temperature (RT). Their calculations show that ground state structure of $\text{Li}_6.25\text{PS}_5(\text{BH}_4)_{0.75}$ has certain orientational disorder of BH_4^- and, in $\text{Li}_6\text{POS}_4(\text{SH})$, SH^- units forming disordered rotor phase upon thermal excitation. They also show rotations of the cluster induce compressing or stretching deformation in each anion-Li pair through MD simulation. In aspect of energy barrier for lithium conduction, total interaction potential forms misalignment of minima of each energy and shows multiple local minima around original lithium site for different deformation and different anion. Based on these results, they provide the mechanistic theory to estimate the activation barrier in superionic conduction. The theory explains that energy barrier for lithium conduction in ionic conductor is determined by two factors: (1) The number of anions interacting with lithium locally (the fewer, the smaller), (2) Extent of the separation of the potential components by inter-atomic deformation. About the effect of inter-atomic deformation, they conclude that deformation raises energy of associated Li sites and reduce the migration barrier and, degenerated deformation states are available high-energy states of Li-ion and form prolonged mean lifetime with ultra-high ionic conductivity. This manuscript is well-written with interesting mechanistic rules that may allow the researchers to design new solid electrolyte materials and to understand superionic conduction with disordered anion clusters; however, I have a number of concerns about the manuscript that need addressing in details before publication.

1) The authors showed that $\text{Li}_6\text{POS}_4(\text{SH})$ and $\text{Li}_6.25\text{PS}_5.25(\text{BH}_4)_{0.75}$ have superionic behavior with weak correlation between Li ions compared to LGPS, LLZO, and LATP (Fig. S4). They mentioned in the introduction section that the superionic conductors have characteristic of collective motion with strong interactions between Li ions, but, didn't explain why $\text{Li}_6\text{POS}_4(\text{SH})$ and $\text{Li}_6.25\text{PS}_5.25(\text{BH}_4)_{0.75}$ have high ionic conductivity without strong collective mode. Please discuss about this.

2) The authors explained that the larger the magnitude of the deformation induces the smaller activation energy. Is there optimal point of the extent of deformation (Fig. 4B)? Also please plot it for more deformation.

3) The authors explained the mechanistic theory using simple two-Li model system. This model system seems not to reflect the cation's effect. When consider the cation, do you expect it shows similar tendency? Is this theory that can be applied to other structural types of solid electrolytes?

4) The authors showed the probability distribution function of the deformation (Fig. 4C). But it is hard to clearly compare two materials without the reference. Please compare these two materials with $\text{Li}_6.25\text{PS}_5.25\text{ClO}_7.75$. Also plot them with same x axis scales.

5) There is no explanation about Figs. 4E and 4F in main text. You should explain them in the main text.

Reviewer #2 (Remarks to the Author):

The manuscript by Fang and Jena reports the computational discovery of two new Li ion conductors with high predicted Li conductivity and potential use for solid-state batteries. The authors also propose a model/theory that might explain the high ionic conductivity of the materials and related compounds.

The general topic, the discovery of electrolytes for solid-state batteries, is timely, and the manuscript is mostly well written. The first part of the paper that discusses the computational discovery and characterization of the two compounds is technically state of the art, and the results are convincing. An experimental validation would be better, but the computational analysis is thorough. The second part of the manuscript, the development and discussion of the model, is quite technical, overly complicated, and not all conclusions are, in my opinion, justified. The developed theory is not as general as stated in the abstract and is not properly put into the context of prior work. Overall, the theory also does not add much new insight to the understanding of superionic conductors.

As such, I cannot recommend the manuscript for publication in its current form, and at least a major revision of the manuscript will be needed. Given the highly technical nature of the second part of the manuscript, a different journal might also be a better venue.

1. Novelty and engagement with prior work

Many theories for ionic conductivity have been proposed in the literature since the 1960s and have been refined in recent years. Despite the rich literature in the field, this article only cites a few examples in the introduction and does not even fully compare the proposed theory to the cited publications. In a revision, the engagement with prior work should be vastly extended.

The main conclusion of the present article is essentially that large anions can result in less specific anion-Li interactions and may create degenerate Li sites that individually have low site energies. The idea that degenerate sites and disorder are beneficial for ionic conduction dates back to the 1970s [1]. The impact of thermal energy and the vibrational free energy in this regard has also been considered early [2]. Related to this early work, the key realization of the cited reference 15 (Wang et al.) is that the BCC anion framework creates degenerate cation sites and no other interstitial sites that could become more stable (in contrast to, e.g., FCC, which exhibits octahedral and tetrahedral sites). This is currently misrepresented in the manuscript. The impact of disorder and site degeneracy has been extensively discussed in more recent literature, especially in the context of the LLZO solid electrolyte and related materials. The impact of thermal motion and distortions of the static sublattice has also been subject of many studies, e.g., recent computational work by Smith and Siegel for highly related materials [3]. There is a significant body of recent literature on the theory of Li-ion conductors (e.g., recent work by Mo and coworkers, Reed and coworkers, Marzari and coworkers). What are the true novel aspects of the proposed theory? Is the model in agreement or in contradiction with prior work? The discussion needs to be significantly extended and corrected. And the statements of generality in the abstract and conclusion should be revised to reflect the fact that the model only deals with one specific mechanism of enhancing ionic conductivity.

[1] Armstrong et al., *J. Solid State Chem.* 8, 1973, 219; [https://doi.org/10.1016/0022-4596\(73\)90088-1](https://doi.org/10.1016/0022-4596(73)90088-1)

[2] O'Reilly, *pssb* 48, 1978, 489-496; <https://doi.org/10.1002/pssa.2210480228>

[3] Smith and Siegel, *Nat. Commun.* 11, 2020, 1483; <https://doi.org/10.1038/s41467-020-15245-5>

2. Validation of the potential

The pair-potential model introduced in the manuscript is only indirectly validated, although a direct comparison with first principles should be possible. How do activation energies for Li escape from DFT compare to the potential model? How does the potential energy of the Li-anion bond compare? DFT calculations could be performed for distorted structures obtained from MD simulations.

Reviewer #3 (Remarks to the Author):

Fang and Jena report computational discovery of two Li superionic conductors. This is an area actively researched for solid-state batteries, and new materials can have a big impact. The reported conductivities look promising with barriers within 0.1-0.2 eV. The manuscript is well-written. I have two main concerns study:

1) Authors start with the chemical composition of Argyrodite family; $\text{Li}_6\text{PS}_5\text{X}$, and try two polyanions in place of X: OH^- and BH_4^- , which is a plausible strategy. They then carry out crystal structure prediction using particle swarm optimization, and find the (nearly) stable, unique crystal structures for both compositions. Unfortunately, while relative stability is a good indicator, it is far from providing any guarantee of synthesizability of a certain crystal structure. This reduces the remainder of the paper to a discussion over hypothetical compounds, rather than some compounds that are derived from Argyrodite family. This, in my opinion, would make the paper more suitable for a more specialized journal.

2) The mechanistic theory part is long and confusing. The model is constructed as pair-wise Coulomb-type, allowing position deviations from minima as a main variable, along with an interaction strength parameter k . This could have been fine for making a point on how distortions of Li-anion pairs can facilitate diffusion. But authors provide lengthy analyses and discussions, and claim that mechanistic theory sheds light into the diffusion mechanism. But the model basically has no degree of freedom to show anything other than local deformations and interaction strength as responsible for what's observed. This sounds like circular logic to me. Even vacancy formation on anion site is claimed to be captured by the model by $k=0$ and reducing "average" k and in turn activation energy. I find this confusing, as it would depend on the overall diffusion mechanism/pathway established in presence of vacancies, not just "averaging". Authors have plenty of AIMD data to look and find what actually is happening mechanistically that is facilitating the diffusion, rather than the need for indirect discussions through a simplified model.

Some minor points:

- Did authors use a particular library for PSO, or did they write their own code?
- The mechanisms of Li diffusion in SSE's is also actively researched -- what's newly uncovered by authors?

Response Letter

Argyrodite-Type Advanced Lithium Conductors and Transport Mechanisms beyond Peddle-wheel Effect

(NCOMMS-21-18521)

Hong Fang and Puru Jena

Department of Physics, Virginia Commonwealth University, Richmond, VA 23238, USA

We are grateful to the reviewers for their insightful, constructive, and detailed review. We thank them for taking the time and helping us to improve the paper. As reflected in the comments from all reviewers, the major concern is focused on the mechanism section of the original paper. Following the reviewer's advice, instead of trying to explain the lithium conductors using an abstract phenomenological model, we have conducted a new systematic study of the ionic diffusions in the materials, entirely based on the molecular dynamics simulation data as well as DFT calculations of local pathways. From the study, we obtain a clear picture of the ionic conduction of the materials, uncovering several new ionic diffusion mechanisms, including:

- (1) A long-range ionic conduction can be achieved by a relay mechanism between local and lattice-crossing diffusions caused by Li-Li repulsions. This is featured as the 'billiard-ball' mechanism. It shows that the local diffusion within the sulfur-block of the argyrodite structure can significantly contribute to the lattice-crossing ionic conduction via the following processes: (a) A long-range diffusion can be triggered by a local diffusion which in turn triggers other local diffusions during its lattice-crossing migration; (b) A long-range diffusion event can start itself due to thermal excitation, triggering other local diffusions along its trajectory; (c) A local diffusion, triggered by a lattice-crossing diffusion, can further trigger other local diffusions in its vicinity.

- (2) There is also a relay mechanism of pure local diffusions due to the Li-Li repulsion. This is featured as the 'revolving-door' mechanism. Although interesting, the diffusions due to this mechanism cannot contribute to the long-range ionic conduction.
- (3) Another interesting mechanism is the 'docking-undocking' of the Li-ion at a cluster. A Li-ion, initially out of any sulfur-blocks in the argyrodite structure, can later be released by a cluster to a sulfur-block and participate in diffusions inside a sulfur-block, or it can be trapped at the end of a lattice-crossing diffusion and stay away from any diffusion event for a significant amount of time (> 30 pico-seconds).

We have also elucidated the paddle-wheel effect in the current context with light mono-anions as compared to much heavier polyanions in the lithium conductors before. A mono-anion cluster (e.g., SH^- or BH_4^- in this case) with low-charge states on its vertices favors weak local interactions in a large crystalline framework (e.g., the argyrodite structure). This can be advantageous, since it allows the cluster to have exceptional (rotational and translational) degrees of freedom in the lattice, so that it has high ability to lower the pathway energy of a passing Li-ion using its rotational and translational dynamics. This is featured as the effect of 'responsive' correlation. On the other hand, the cluster with excessive degrees of freedom will sustain great (rotational and translational) dynamics under thermal excitation. It is shown that such 'active' dynamics may interrupt the 'responsive' correlation above and hinder the Li-ion from migration by inadvertently raising the pathway energy. Thus, to achieve the highest ionic conductivity at low temperature, it is important to reach an optimized cluster-dynamics in a system.

Based on these new results, we rewrote the entire 'mechanism' section in the revised paper and presented a complete picture of the ionic diffusion and conduction as well as the correlation of cluster and Li-ion dynamics. Since the paper has been substantially revised, we sincerely ask the reviewers to refer to the whole paper. The revised and the added parts in the revised paper are highlighted in blue. Following is our point-by-point response to the reviewers' comments.

Reviewer #1 (Remarks to the Author):

This manuscript presents the computational study on a universal theory of the principle and mechanism of superionic lithium-ion conduction at atomic scale and suggests new argyrodite-type solid electrolytes which have very high ionic conductivity. Through ab initio molecular dynamics (AIMD) simulations, they show that $\text{Li}_6.25\text{PS}_5(\text{BH}_4)_{0.75}$ and $\text{Li}_6\text{POS}_4(\text{SH})$ have super-high ionic conductivities of 177 and 82 mS/cm with low activation energies of 0.108eV and 0.166eV, respectively, at room temperature (RT). Their calculations show that ground state structure of $\text{Li}_6.25\text{PS}_5(\text{BH}_4)_{0.75}$ has certain orientational disorder of BH_4^- and, in $\text{Li}_6\text{POS}_4(\text{SH})$, SH^- units forming disordered rotor phase upon thermal excitation. They also show rotations of the cluster induce compressing or stretching deformation in each anion-Li pair through MD simulation. In aspect of energy barrier for lithium conduction, total interaction potential forms misalignment of minima of each energy and shows multiple local minima around original lithium site for different deformation and different anion. Based on these results, they provide the mechanistic theory to estimate the activation barrier in superionic conduction. The theory explains that energy barrier for lithium conduction in ionic conductor is determined by two factors: (1) The number of anions interacting with lithium locally (the fewer, the smaller), (2) Extent of the separation of the potential components by inter-atomic deformation. About the effect of inter-atomic deformation, they conclude that deformation raises energy of associated Li sites and reduce the migration barrier and, degenerated deformation states are available high-energy states of Li-ion and form prolonged mean lifetime with ultra-high ionic conductivity. This manuscript is well-written with interesting mechanistic rules that may allow the researchers to design new solid electrolyte materials and to understand superionic conduction with disordered anion clusters; however, I have a number of concerns about the manuscript that need addressing in details before publication.

Reviewer's Comment 1:

1) The authors showed that $\text{Li}_6\text{POS}_4(\text{SH})$ and $\text{Li}_6.25\text{PS}_5.25(\text{BH}_4)_{0.75}$ have superionic behavior with weak correlation between Li ions compared to LGPS, LLZO, and LATP (Fig. S4). They mentioned in the introduction section that the superionic conductors have characteristic of collective motion with strong interactions between Li ions, but, didn't

explain why $\text{Li}_6\text{POS}_4(\text{SH})$ and $\text{Li}_{6.25}\text{PS}_{5.25}(\text{BH}_4)_{0.75}$ have high ionic conductivity without strong collective mode. Please discuss about this.

Author Response:

We thank the reviewer for the detailed review and the insightful comment. Collective motion is one of a few conduction mechanisms that is responsible for fast-ion diffusions in some lithium conductor types. In the current materials with the argyrodite structure (i.e., $\text{Li}_6\text{POS}_4(\text{SH})$ and $\text{Li}_{6.25}\text{PS}_{5.25}(\text{BH}_4)_{0.75}$), according to our study in the revised paper, the long-range ionic conduction is contributed by individual large-displacement diffusions, lattice-crossing diffusions formed by a relay mechanism between local and long-distance diffusions due to Li-Li repulsions (featured as the 'billiard-ball' mechanism), as well as the correlational dynamics between the Li-ion and the clusters. Details of these, including our methodology to obtain the results, are given in the 'Results and Discussion' section on page 5-24 of the revised paper.

Reviewer's Comment 2:

2) The authors explained that the larger the magnitude of the deformation induces the smaller activation energy. Is there optimal point of the extent of deformation (Fig. 4B)? Also please plot it for more deformation.

Author Response:

We thank the reviewer for the question. As mentioned at the beginning of this letter, following the reviewers' advice, we have conducted new studies on the ionic diffusion mechanisms of the materials based on the molecular dynamics simulations and DFT pathway calculations, rather than using a phenomenological model like in the original paper. Therefore, we now provide a direct and clear physical picture of the ionic diffusions and the cluster dynamics of the materials, rather than relying on the 'local Li-anion' deformations in the phenomenological model. Specifically, we first identify all the long-live large-displacement diffusion events of the studied materials at low temperature (300 K), based on the molecular dynamics simulation data. Then, after a detailed analysis of all the events and their related ones, we combine the individual events in time and space. By

studying these combined events, we obtain a complete picture of the ionic diffusions in these systems with several interesting ionic-diffusion mechanisms uncovered. Thirdly, based on the knowledge gained from studying the cluster dynamics and their correlations with the Li-ion displacement, we further design a set of DFT calculations (using selective dynamics and nudged elastic band method) to study the pathway energies of the Li-ion migration and elucidate the correlational dynamics between the Li-ion and the cluster. Please refer to page 11-24 in the revised paper, especially the discussions about Figure 3-6, as well as the discussion on page 25-27 in the ‘Conclusion’ section.

Reviewer’s Comment 3:

3) The authors explained the mechanistic theory using simple two-Li model system. This model system seems not to reflect the cation’s effect. When consider the cation, do you expect it shows similar tendency? Is this theory that can be applied to other structural types of solid electrolytes?

Author Response:

We thank the reviewer for the insightful questions. As mentioned in the previous response, we have conducted a new mechanistic study of the materials in the revised paper. This study provides us with a complete picture of the Li-ion diffusions due to the Li-anion (cluster) as well as Li-Li interactions. It is found that the Li-Li repulsion interaction plays a key role in the ionic diffusions of the materials, as evidenced by the currently found ‘billiard ball’ mechanism as well as the ‘revolving door’ mechanism described in the revised paper (e.g., the discussion on page 17-18). On the other hand, the Li-anion(cluster) interaction is the other key part of the ionic diffusion in the materials, as evidenced by the ‘docking-undocking’ mechanism described in the revised paper (e.g., the discussion on page 18) as well as the important Li-cluster correlational dynamics which can lower the Li-ion pathway energy (e.g., the discussion on page 19-24). The mechanisms and the correlational effect are particularly useful for cluster-containing ionic conductors (including the ones with structures other than the current argyrodite structure), especially,

with small mono-anion clusters, as discussed in the second paragraph on page 10 and the second paragraph on page 18.

Reviewer's Comment 4:

4) The authors showed the probability distribution function of the deformation (Fig. 4C). But it is hard to clearly compare two materials without the reference. Please compare these two materials with $\text{Li}_6.25\text{PS}_{5.25}\text{Cl}_{0.75}$. Also plot them with same x axis scales.

Author Response:

We thank the reviewer for this question. As discussed in the previous responses, we have conducted a new mechanistic study of the materials in the revised paper based solely on the molecular dynamics simulation data and DFT local pathway calculations. Realizing that keeping the original phenomenological model in the revised paper would only lead to confusion, we removed the phenomenological model, including the probability distribution of the deformation, from the paper.

Reviewer's Comment 5:

5) There is no explanation about Figs. 4E and 4F in main text. You should explain them in the main text.

Author Response:

We thank the reviewer for the question. The original Fig. 4E and 4F (as shown below) is to find out all the interstitial sites that can accommodate at least the size of one Li-ion in different structures. Fig. 4E is a schematic version to show this and Fig. 4F shows the total volumes of such interstitial sites available in different materials (as put in different line colors). The solid lines are for all the interstitial sites surrounding each Li-ion in a material as depicted by the shaded-circle in Fig. E, while the dotted lines are for the

interstitial sites in-between two neighboring Li-ions as depicted by the overlapping part of the circles in Fig. E.

As mentioned in the above responses, we have conducted a new study using new methodologies and have removed this figure from the paper. However, in the revised paper, we have conducted a similar interstitial analysis for the studied structures, to gain knowledge on the minimum diffusion distance of a material, as described on page 12 and in Figure S6 of SI.

Reviewer #2 (Remarks to the Author):

The manuscript by Fang and Jena reports the computational discovery of two new Li ion conductors with high predicted Li conductivity and potential use for solid-state batteries. The authors also propose a model/theory that might explain the high ionic conductivity of the materials and related compounds.

The general topic, the discovery of electrolytes for solid-state batteries, is timely, and the manuscript is mostly well written. The first part of the paper that discusses the computational discovery and characterization of the two compounds is technically state of the art, and the results are convincing. An experimental validation would be better, but the computational analysis is thorough. The second part of the manuscript, the development and discussion of the model, is quite technical, overly complicated, and not all conclusions are, in my opinion, justified. The developed theory is not as general as stated in the abstract and is not properly put into the context of prior work. Overall, the theory also does not add much new insight to the understanding of superionic conductors.

As such, I cannot recommend the manuscript for publication in its current form, and at least a major revision of the manuscript will be needed. Given the highly technical nature of the second part of the manuscript, a different journal might also be a better venue.

Reviewer's Comment 1:

1. Novelty and engagement with prior work

Many theories for ionic conductivity have been proposed in the literature since the 1960s and have been refined in recent years. Despite the rich literature in the field, this article only cites a few examples in the introduction and does not even fully compare the proposed theory to the cited publications. In a revision, the engagement with prior work should be vastly extended.

The main conclusion of the present article is essentially that large anions can result in less specific anion-Li interactions and may create degenerate Li sites that individually have low

site energies. The idea that degenerate sites and disorder are beneficial for ionic conduction dates back to the 1970s [1]. The impact of thermal energy and the vibrational free energy in this regard has also been considered early [2]. Related to this early work, the key realization of the cited reference 15 (Wang et al.) is that the BCC anion framework creates degenerate cation sites and no other interstitial sites that could become more stable (in contrast to, e.g., FCC, which exhibits octahedral and tetrahedral sites). This is currently misrepresented in the manuscript. The impact of disorder and site degeneracy has been extensively discussed in more recent literature, especially in the context of the LLZO solid electrolyte and related materials. The impact of thermal motion and distortions of the static sublattice has also been subject of many studies, e.g., recent computational work by Smith and Siegel for highly related materials [3]. There is a significant body of recent literature on the theory of Li-ion conductors (e.g., recent work by Mo and coworkers, Reed and coworkers, Marzari and coworkers). What are the true novel aspects of the proposed theory? Is the model in agreement or in contradiction with prior work? The discussion needs to be significantly extended and corrected. And the statements of generality in the abstract and conclusion should be revised to reflect the fact that the model only deals with one specific mechanism of enhancing ionic conductivity.

[1] Armstrong et al., J. Solid State Chem. 8, 1973, 219; [https://doi.org/10.1016/0022-4596\(73\)90088-1](https://doi.org/10.1016/0022-4596(73)90088-1)

[2] O'Reilly, pssb 48, 1978, 489-496; <https://doi.org/10.1002/pssa.2210480228>

[3] Smith and Siegel, Nat. Commun. 11, 2020, 1483; <https://doi.org/10.1038/s41467-020-15245-5>.

Author Response:

We are grateful to the reviewer for constructive and detailed comments and advice. We find these to be very helpful. Consequently, we have conducted new studies on the ionic diffusions of the materials and uncovered several interesting diffusion mechanisms.

In addition, we elucidate the ‘paddle-wheel’ effect in the current context with small mono-anion clusters.

To address the reviewer’s concern on presenting the previous works and putting our current study in context, we have added a new section in the Introduction of the revised paper by reviewing the main ionic conduction mechanisms in modern lithium superionic conductors, especially reviewing the works related to the cluster-containing lithium conductors with the peddle-wheel effect. This new part is on page 2-4 of the revised paper, with new references added as Ref. 15-34, including the ones mentioned by the reviewer:

Ref. 15. Armstrong, R. D.; Bulmer, R. S.; Dickinson, T., Some Factors Responsible for High Ionic Conductivity in Simple Solid Compounds, *J. Sol. Stat. Chem.* 8, 219-228, 1973.

Ref. 21. Smith, J. G.; Siegel, D. J. Low-temperature paddlewheel effect in glassy solid electrolytes. *Nature Communications* 11, 1483, 2020.

With the background knowledge provided, we then point out the difference of the current study, in terms of incorporating small (light) mono-anion clusters (e.g., BH_4^- and SH^-) into a modern lithium conductor structure (e.g., argyrodite structure). Please refer to the added part on page 3-5 of the revised paper.

Instead of relying on the original phenomenological model, we have conducted a systematic study on the ionic diffusions in the materials, entirely based on the molecular dynamics simulation data as well as DFT calculations of local pathways. This addresses the reviewer’s question about the novel aspects/mechanisms gained from the current study. One of the main methods we used is motivated by the one used in the newly cited Ref. 21 mentioned by the reviewer. From our new study, we obtain a clear picture of the ionic diffusions in the materials, with several new ionic diffusion mechanisms revealed. These include:

First, it is found that a long-range ionic conduction can be achieved by a relay mechanism between the local and lattice-crossing diffusions due to Li-Li repulsions. This is referred to as the 'billiard-ball' mechanism. Before, it was believed that local Li-ion diffusions within a sulfur-block will not contribute to the long-range ionic conduction in

the argyrodite structure. Here, instead, we find that the local diffusion within the sulfur-block can significantly contribute to the lattice-crossing ionic conduction via the following processes: (a) A long-range diffusion can be triggered by a local diffusion and then it can trigger local diffusions in other sulfur-blocks during its lattice-crossing migration; (b) A long-range diffusion can start itself due to thermal excitation and trigger other local diffusions along its trajectory; (c) A local diffusion, triggered by a lattice-crossing diffusion, can trigger other local diffusions in its vicinity.

Second, a relay of local diffusions due to the Li-Li repulsion, referred to as the 'revolving-door' mechanism, is also uncovered. Although interesting, the diffusions due to this mechanism cannot contribute to the long-range ionic conduction. Another interesting mechanism observed is the 'docking-undocking' of Li-ion at a cluster. A Li-ion, initially out of any sulfur-blocks in the argyrodite structure, can later be released by a cluster to a sulfur-block and participate in diffusions inside a sulfur-block, or it can be trapped at the end of a lattice-crossing diffusion and stay away from any diffusion event for a significant amount of time (> 30 pico-seconds).

Third, in the revised paper, we have elucidated the paddle-wheel effect in the current context with light mono-anions as compared to much heavier polyanions in the lithium conductors before. A mono-anion cluster (e.g., SH^- or BH_4^- in this case) with low-charge states on its vertices favors weak local interactions in a large crystalline framework (e.g., the argyrodite structure). This can be advantageous, since it allows the cluster to have exceptional (rotational and translational) degrees of freedom in the lattice, so that it has high ability to lower the pathway energy of a passing Li-ion using its rotational and translational dynamics. This is referred to as the effect of 'responsive' correlation. On the other hand, the cluster with excess degrees of freedom will sustain great (rotational and translational) dynamics under thermal excitation. It is shown that such 'active' dynamics may interrupt the 'responsive' correlation above and hinder the Li-ion migration by inadvertently raising the pathway energy. Thus, to achieve the highest ionic conductivity at low temperature, it is important to reach an optimized cluster-dynamics in a system.

Please refer to page 10-20 (with Fig. 4-5) in the revised paper for a detailed discussion on the “First” and “Second” points above. Please refer to page 20-24 for a discussion on the “Third” point above.

We have changed the Title and the Abstract according to the scope of the revised paper.

Reviewer’s Comment 2:

2. Validation of the potential

The pair-potential model introduced in the manuscript is only indirectly validated, although a direct comparison with first principles should be possible. How do activation energies for Li escape from DFT compare to the potential model? How does the potential energy of the Li-anion bond compare? DFT calculations could be performed for distorted structures obtained from MD simulations.

Author Response:

We thank the reviewer for the comments. To answer these questions, we have followed the reviewer’s advice and conducted a set of DFT calculations to study the Li-ion migration pathways. Specifically, to understand the correlation between the rotational dynamics of the cluster and the Li-ion transport in the materials, we carried out a set of DFT calculations on the local migration pathways using selective dynamics and nudged elastic band method. From the study, we found how the responsive rotations of the cluster can facilitate the migration of the Li-ion, and how the ‘active’ rotations (e.g., due to thermal excitation) of the cluster may change the energy pathways. These results are useful to understand the paddle-wheel effect in the current context with small mono-anion clusters SH^- and BH_4^- . Please refer to the discussions on page 20-24 (with Fig. 6) of the revised paper.

Reviewer #3 (Remarks to the Author):

Fang and Jena report computational discovery of two Li superionic conductors. This is an area actively researched for solid-state batteries, and new materials can have a big impact. The reported conductivities look promising with barriers within 0.1-0.2 eV. The manuscript is well-written. I have two main concerns study:

Reviewer's Comment 1:

1) Authors start with the chemical composition of Argyrodite family; $\text{Li}_6\text{PS}_5\text{X}$, and try two polyanions in place of X: OH^- and BH_4^- , which is a plausible strategy. They then carry out crystal structure prediction using particle swarm optimization, and find the (nearly) stable, unique crystal structures for both compositions. Unfortunately, while relative stability is a good indicator, it is far from providing any guarantee of synthesizability of a certain crystal structure. This reduces the remainder of the paper to a discussion over hypothetical compounds, rather than some compounds that are derived from Argyrodite family. This, in my opinion, would make the paper more suitable for a more specialized journal.

Author Response:

We thank the reviewer for the comment. Although these materials have not been fully synthesized, there is some promising indicators to realize them in the future, as pointed out in the revised paper. The used mono-anion clusters, OH^- for $\text{Li}_6\text{PS}_5\text{OH}/\text{Li}_6\text{POS}_4(\text{SH})$ and BH_4^- for $\text{Li}_{6.25}\text{PS}_{5.25}(\text{BH}_4)_{0.75}$, are readily available in practice. Also, they have already been used to successfully synthesize novel bulk materials, including lithium superionic conductors, such as the complex hydride perovskites described in Ref. 39 and the well-known $\text{Li}_2(\text{OH})\text{X}$ ($\text{X} = \text{halogen}$) lithium conductors in Ref. 40. Recently, a glassy matrix of $(100-x) \cdot (0.75\text{Li}_2\text{S} \cdot 0.25\text{P}_2\text{S}_5) \cdot x\text{LiBH}_4$ containing a crystalline phase of $\text{Li}_5\text{PS}_4(\text{BH}_4)_2$ (with excess BH_4) has been successfully prepared by a mechanochemical process, with the BH_4^- clusters found at the halogen sites as in the

lithium argyrodites (Ref. 41 in the paper). We have pointed these out in the last paragraph on page 26 of the revised paper.

More importantly, since it is found that these materials can entail very high ionic conductivities at room temperature and exceptionally low activation energies, by studying their ionic transport behaviors together with the cluster dynamics, some new fundamental ionic diffusion mechanisms can be uncovered. This will be helpful for the design of new and powerful lithium superionic conductors, especially those that contain anion clusters featured with cluster dynamics. This is evidenced by several new ionic diffusion mechanisms as well as the interesting Li-cluster correlational dynamics presented in the revised paper.

Reviewer's Comment 2:

2) The mechanistic theory part is long and confusing. The model is constructed as pairwise Coulomb-type, allowing position deviations from minima as a main variable, along with an interaction strength parameter k . This could have been fine for making a point on how distortions of Li-anion pairs can facilitate diffusion. But authors provide lengthy analyses and discussions, and claim that mechanistic theory sheds light into the diffusion mechanism. But the model basically has no degree of freedom to show anything other than local deformations and interaction strength as responsible for what's observed. This sounds like circular logic to me. Even vacancy formation on anion site is claimed to be captured by the model by $k=0$ and reducing "average" k and in turn activation energy. I find this confusing, as it would depend on the overall diffusion mechanism/pathway established in presence of vacancies, not just "averaging". Authors have plenty of AIMD data to look and find what actually is happening mechanistically that is facilitating the diffusion, rather than the need for indirect discussions through a simplified model.

Author Response:

We thank the reviewer for the constructive and insightful advice. Following the advice, we have conducted a new systematic study of the ionic diffusions in the materials,

which is entirely based on the molecular dynamics simulation data and the DFT pathway calculations (as stated at the beginning of this letter). We have rewritten the second half of the paper, removing the whole section on the phenomenological model. Please refer to page 10-24 of the revised paper for the detailed discussions. Please also refer to page 25-26 of the revised paper, where the new insights gained from the study are summarized and discussed.

Reviewer's Comment 3:

Some minor points:

- Did authors use a particular library for PSO, or did they write their own code?

Author Response:

We used the library of CALYPSO package for PSO. We have added this in the Method section (page 27) of the revised paper with the reference to the package added as Ref. 42.

Reviewer's Comment 4:

- The mechanisms of Li diffusion in SSE's is also actively researched -- what's newly uncovered by authors?

Author Response:

To address the reviewer's concern, as stated previously, we have carried out a systematic study of the ionic transport mechanisms of the material. Following is a summary of what are newly uncovered from the study. Please also refer to the whole paper for the context of the study and the detailed discussions.

First, it is found that a long-range ionic conduction can be achieved by a relay mechanism between local and lattice-crossing diffusions due to Li-Li repulsions. This is referred to as the 'billiard-ball' mechanism. It was believed that local Li-ion diffusions inside a sulfur-block cannot contribute to any long-range ionic conduction in the argyrodite. Here, however, we find that the local diffusion within the sulfur-block can significantly contribute to the lattice-crossing ionic conduction via the following behaviors: (a) A long-range diffusion can be triggered by a local diffusion and then it can trigger local diffusions in other sulfur-blocks; (b) A long-range diffusion event can start itself due to thermal excitation and trigger local diffusions in other sulfur blocks along its trajectory; (c) A local diffusion, triggered by a lattice-crossing diffusion, can trigger other local diffusions in its vicinity.

Second, a relay of local diffusions due to the Li-Li repulsion, featured as the 'revolving-door' mechanism, is also uncovered. Although interesting, the diffusions due to this mechanism cannot contribute to the long-range ionic conduction. Another interesting mechanism observed is the 'docking-undocking' of Li-ion at a cluster. A Li-ion, initially out of any sulfur-blocks in the argyrodite structure, can later be released by a cluster to a sulfur-block and participate in diffusions inside a sulfur-block, or it can be trapped at the end of a lattice-crossing diffusion and stay away from any diffusion event for a significant amount of time (> 30 pico-seconds).

Third, in the revised paper, we have elucidated the so-called paddle-wheel effect in the current context with light mono-anions as compared to much heavier polyanions in the lithium conductors before. A mono-anion cluster (e.g., SH^- or BH_4^- in this case) with low-charge states on its vertices is in favor of weak local interactions in a large crystalline framework (e.g., the argyrodite structure). This can be advantageous on the one hand, since it allows the cluster to have exceptional (rotational and translational) degrees of freedom in the lattice, so that it has high ability to lower the pathway energy of a passing Li-ion by interacting with the Li-ion, using its rotational and translational dynamics. This is featured as the effect of 'responsive' correlation. On the other hand, the cluster with excessive degrees of freedom will sustain great (rotational and translational) dynamics under thermal excitation. It is shown that such 'active' dynamics may interrupt the 'responsive' correlation above and hinder the Li-ion migration by inadvertently raise the pathway energy. Thus, to

achieve the highest ionic conductivity at low temperature, it is important to reach an optimized cluster-dynamics in a system.

REVIEWER COMMENTS

Reviewer #1 (Remarks to the Author):

The authors have largely revised the manuscript with further analyses and discussion on the diffusion mechanism. They pointed that the paddle-wheel effect exists in both $\text{Li}_6.25\text{PS}_5.25(\text{BH}_4)_{0.75}$ and $\text{Li}_6\text{POS}_4(\text{SH})$, and, explained three diffusion types based on their AIMD simulation results with different Events, 1) 'billiard-ball' mechanism : the intra-diffusion trigger the inter-diffusion (or the inter-diffusion trigger the intra-diffusion), 2) 'revolving-door' mechanism : the intra-diffusion without inter-diffusion induced by Li-Li repulsion, and 3) 'docking-undocking' mechanism : when the Li-ion strongly interact with an anion cluster, the Li-ion trapped over tens of pico-seconds. This revised manuscript has been improved with better explanation on the diffusion mechanism such as the paddle wheel effect than previous one. However, some issues are still not clearly addressed. The main concern is that whether these design strategy for the advanced lithium superionic conductors (ALiSIC) can be generally applied to other systems as they emphasized it as their work's novelty. The general applicability of their design strategy for ALiSIC and following issues should be addressed before publication in Nature Communications:

1) In this work, it is very important to predict accurate lithium conductivities of argyrodite compounds with the different chemical composition to propose new superionic conducting materials. It is well known that the predicted lithium conductivity using AIMD simulations varies by one or two orders of magnitude even for same composition and crystal structure with the convergence criteria. This is because AIMD simulations include only a limited number of diffusion events especially at low temperature (e.g. below 600 K), leading to significant statistical variances in calculated lithium conductivities. The authors mentioned "Typical AIMD simulations last over 100ps until the extracted Li-ion diffusion coefficients are converged." (in page 28), but, there is no clear explanation on the criteria of the convergence. The authors should explain this with MSD data at low temperature and how the room temperature conductivity changes with running time for AIMD simulation at low temperatures. They may use the relative standard deviation (RSD) of the diffusivities proposed by the previous computational report, [npj Computational Materials, 4, 18 (2018)]. They suggested that AIMD simulations require enough times to observe the ion hops, which are related to the RSD of the diffusivities, and selected RSD of 0.2-0.3 as the criteria of the convergence in another report [Journal of the American Chemical Society, 142, 7012 (2020)].

2) The authors propose the design strategy of ALiSIC by incorporating small mono-anion clusters into a modern fast-ion framework. They show that the substitution of the mono-anion cluster (SH- or BH4-) to lithium argyrodite ($\text{Li}_6\text{PS}_5\text{X}$) makes them ALiSIC with RT ionic conductivity ~ 0.1 S/cm and activation energies ~ 0.1 eV. I don't think that this strategy can be applied to the known fast-ion frameworks such as LGPS, garnet-LLZO, NASICON, and perovskite-LLTO as they do not have halogen sites that can be substituted by the mono-anion cluster. More detailed discussion on the applicable

fast-ion framework in addition to lithium argyrodite family should be provided to claim that this strategy is powerful.

3) In page 11, the statement, "... with small rotational barriers of SH⁻ and BH₄⁻ estimated to be about 30 and 10 meV, respectively.", was written. How to estimate the rotational barriers of the clusters?

4) The authors separate cluster rotation types, thermally-excited rotation ('active' rotation), and the rotation interacted with Li-ion ('responsive' rotation). To confirm these effects, they conducted NEB calculations with/without other atomic motions by using selective dynamics. But, the method to confirm rotation types, especially 'active' rotation, is confusing (Fig. 6). Please discuss this.

5) Fig. 5: The Event numbers are wrong (1,2,3,4 should be 4,5,6,7)

6) Fig. 6: In the plot of migration barriers, there are no explanations about abbreviations such as 'F', 'M', 'PM', and 'Ext'. Please demonstrate them.

7) In page 24, the statement, "... as shown in Fig. 6F, the energy barriers of both P4 and P3 will not be affected by the 'active' rotations ..." was written, but, in Fig. 6F, 'P4' doesn't exist. Please check this.

Reviewer #2 (Remarks to the Author):

In their revision, the authors made substantial changes. In fact, only 4 (!) of the first 25 pages of the original manuscript appear to be the same, all other pages have been completely rewritten. I have to say that this is quite unusual even for a major revision.

However, the manuscript has indeed improved in the revision. Especially the discussion of prior work is now much more thorough and better organized. Nevertheless, the analysis of the Li diffusion mechanism, although completely redone, remains overly complex and could be shortened substantially. This is also the case for the conclusion section, which could be shortened by ~50% and should focus on the take-home messages.

I suggest the authors revise the manuscript once again, this time for conciseness. I recommend to focus on the most important questions, for example, to work out clearly: (i) What is the origin of fast long-range Li conduction in the materials? (ii) In which way is the mechanism similar to or different from previous reports (e.g., paddle-wheel mechanism)? (iii) Can the insight be used to formulate a general design criterion? These points are still not clear.

An important novel insight is (in my opinion) the impact of Li repulsion on the triggering of long-range diffusion (i.e., the "billiard ball mechanism"). Why is this mechanism relevant only for cluster ions? Many Li-ion conductors exhibit fast local Li diffusion that does not contribute to overall Li conductivity. Should the mechanism not generally be of importance for such materials?

Reviewer #3 (Remarks to the Author):

Authors have carried out an extensive revision, and the paper is almost entirely rewritten. I don't have any other comments, at least on my end.

Response Letter

Argyrodite-Type Advanced Lithium Conductors and Transport Mechanisms beyond Paddle-wheel Effect

(NCOMMS-21-18521A)

Hong Fang and Puru Jena

Department of Physics, Virginia Commonwealth University, Richmond, VA 23238, USA

We are again grateful to the reviewers for their insightful, constructive, and detailed review. We thank them for taking the time and helping us to further improve the paper. Following the reviewers' advice, we have added new analysis and revised discussions for conciseness and clarity. The added and revised parts in the paper and the Supplemental Information are marked in blue in the “*_marked” versions. Following is our point-by-point response to the reviewers' comments.

Reviewer #1 (Remarks to the Author):

The authors have largely revised the manuscript with further analyses and discussion on the diffusion mechanism. They pointed that the paddle-wheel effect exists in both $\text{Li}_{6.25}\text{PS}_{5.25}(\text{BH}_4)_{0.75}$ and $\text{Li}_6\text{POS}_4(\text{SH})$, and, explained three diffusion types based on their AIMD simulation results with different Events, 1) ‘billiard-ball’ mechanism: the intra-diffusion trigger the inter-diffusion (or the inter-diffusion trigger the intra-diffusion), 2) ‘revolving-door’ mechanism: the intra-diffusion without inter-diffusion induced by Li-Li repulsion, and 3) ‘docking-undocking’ mechanism: when the Li-ion strongly interact with an anion cluster, the Li-ion trapped over tens of pico-seconds. This revised manuscript has been improved with better explanation on the diffusion mechanism such as the paddle wheel effect than previous one. However, some issues are still not clearly addressed. The

main concern is that whether these design strategy for the advanced lithium superionic conductors (ALiSIC) can be generally applied to other systems as they emphasized it as their work's novelty. The general applicability of their design strategy for ALiSIC and following issues should be addressed before publication in Nature Communications:

Reviewer's comment 1:

1) In this work, it is very important to predict accurate lithium conductivities of argyrodite compounds with the different chemical composition to propose new superionic conducting materials. It is well known that the predicted lithium conductivity using AIMD simulations varies by one or two orders of magnitude even for same composition and crystal structure with the convergence criteria. This is because AIMD simulations include only a limited number of diffusion events especially at low temperature (e.g. below 600 K), leading to significant statistical variances in calculated lithium conductivities. The authors mentioned "Typical AIMD simulations last over 100ps until the extracted Li-ion diffusion coefficients are converged." (in page 28), but, there is no clear explanation on the criteria of the convergence. The authors should explain this with MSD data at low temperature and how the room temperature conductivity changes with running time for AIMD simulation at low temperatures. They may use the relative standard deviation (RSD) of the diffusivities proposed by the previous computational report, [npj Computational Materials, 4, 18 (2018)]. They suggested that AIMD simulations require enough times to observe the ion hops, which are related to the RSD of the diffusivities, and selected RSD of 0.2-0.3 as the criteria of the convergence in another report [Journal of the American Chemical Society, 142, 7012 (2020)].

Author's response:

Following the reviewer's advice, we have carried out the statistical variance analysis on our data using the approach introduced in the references mentioned by the reviewer. It is found that the relative standard deviations (RSDs) of our low temperature data are well below the convergence criteria of 0.3. Based on this analysis, we have also included the uncertainties of the calculated room-temperature ionic conductivities of the two studied materials. On page 7 of the revised paper, we added:

“...Since the MD simulations can only include limited number of diffusion events especially at the low temperature, we further conducted a statistical variance analysis on the data using the approach introduced in Ref. 37. As detailed in Section I of the Supplemental Information, the relative standard deviations (RSD) found for the lowest-temperature diffusivities of $\text{Li}_{6.25}\text{PS}_{5.25}(\text{BH}_4)_{0.75}$ (@300 K) and $\text{Li}_6\text{POS}_4(\text{SH})$ (@400K) are 0.235 and 0.052, respectively, well below the convergence criteria of 0.3 [37-38].”

In the revised Supplemental Information, we added a new Section I which contains the statistical variance analysis of our low-temperature data:

“

Section I: Statistical variance analysis of the low-temperature MD data.

Fig. S-I-1 Analysis to determine the lower and upper bounds of the linear fitting to the MSD data.

Following is the statistical variance analysis of the low-temperature MD data using the method introduced in Ref. 37 in the paper:

First, it is found [37] that the linear fit to the mean squared displacement (MSD) (Eq. S1) to obtain the ionic diffusivity (Eq. S2) should be performed on a time range $\Delta t_{\text{low}} \leq t \leq \Delta t_{\text{upper}}$ of the simulation. The data with $t < \Delta t_{\text{low}}$ are due to the harmonic vibration of Li-ions, and the data with $t > \Delta t_{\text{upper}}$ show large deviation due to the poor statistics at large Δt . Therefore, these data should be excluded from the linear fitting. As

shown in the inset of Fig. S-I-1, the value of Δt_{low} is determined to be 4 ps to exclude the so-called ballistic region of the data [37]. To determine Δt_{upper} , four MSD(t) curves (with each up to 50 ps, obtained by dividing a total MD simulation on $\text{Li}_{6.25}\text{PS}_{5.25}(\text{BH}_4)_{0.75}$ at 300 K of 250 ps into four well separated parts) are analyzed. As shown in Fig. SI-1, the goodness of the linear fit, R^2 , with different upper fitting bound, of each MSD(t) curve is evaluated. It is shown that the value of R^2 begins to significantly deviate from 1 after $0.8 \times t_{\text{tot}}$. Therefore, Δt_{upper} is determined to be $0.8 \times t_{\text{tot}}$ in our analysis. Note that the MSD of Li-ions here is calculated from the MD data as,

$$\text{MSD}(\Delta t) = \frac{1}{N} \text{TMSD}(\Delta t), \quad (\text{Eq. S1})$$

where

$$\text{TMSD}(\Delta t) = \sum_{i=1}^N \frac{1}{N_{\Delta t}} \sum_{\Delta t=0}^{t_{\text{tot}}-\Delta t} |r_i(t' + \Delta t) - r_i(t')|^2$$

is the total mean squared displacement and N is the total number of Li-ions in the simulation cell. The diffusivity (D) is obtained from linear fitting to the MSD according to

$$D = \frac{\text{MSD}(\Delta t)}{2d\Delta t} D_{\text{intercept}} \quad (\text{Eq. S2})$$

with $d = 3$ is the dimension of the conduction system.

Second, with the determined Δt_{low} and Δt_{upper} , we went on to evaluate the relative standard deviation (RSD) of the diffusivity, S_d/D_{true} , as defined in Ref. 37. S_d is the standard deviations of a set of fitted D values. These fits are shown in Fig. S-I-2 for $\text{Li}_{6.25}\text{PS}_{5.25}(\text{BH}_4)_{0.75}$ at 300 K up to 100, 110, 120, 130, 140, 150, 180 and 200 ps, and for $\text{Li}_6\text{POS}_4(\text{SH})$ at 400 K up to 80, 90, 100, 110 and 120 ps. D_{true} is calculated from the longest available MD (80% of the total time), which is 200 ps in the case of $\text{Li}_{6.25}\text{PS}_{5.25}(\text{BH}_4)_{0.75}$ at 300 K and 120 ps in the case of $\text{Li}_6\text{POS}_4(\text{SH})$ at 400 K. The calculated RSD is found to be related to the total effective hops, N_{eff} , as [37]

$$\text{RSD} = \frac{A}{\sqrt{N_{\text{eff}}}} + B \quad (\text{Eq. S3})$$

with A and B being the fitting parameters.

Fig. S-I-2 Linear fitting to calculate the RSD of the diffusivity at different time limit or maximum TMSD (Eq. S1) or total effective hops N_{eff} (Eq. S3). The fittings used to obtain the ionic diffusivities at 300 and 400 K (Fig. 1 in the paper) for $Li_{6.25}PS_{5.25}(BH_4)_{0.75}$ and $Li_6POS_4(SH)$, respectively, are put in red, corresponding to 140 and 100 ps.

Here, N_{eff} can be calculated as $\max[TMSD(\Delta t)]/a^2$ [37] with $\max[TMSD(\Delta t)]$ the maximum of TMSD (Eq. S1) over the entire range and $a = 2.0$ and 1.8 \AA for $Li_{6.25}PS_{5.25}(BH_4)_{0.75}$ and $Li_6POS_4(SH)$, respectively, equal to their corresponding threshold diffusion distances in Eq. (1) in the paper. Fig. SI-3 shows the calculated RSD vs. the N_{eff} and fitted by Eq. S3. It is found that the RSD of our calculated diffusivity of $Li_{6.25}PS_{5.25}(BH_4)_{0.75}$ at 300 K is 0.235 (linear fit up to 140 ps). The RSD of our calculated diffusivity of $Li_6POS_4(SH)$ at 400 K is 0.051 (linear fit up to 100 ps). These values are well below the convergence criteria of 0.3 proposed in Ref. 37-38 in the paper.

Fig. S-I-3 Calculated relative standard deviation (RSD) of diffusivity vs. the total effective hops N_{eff} , fitted by Eq. S3. The highlight points marked by the cross correspond to the calculated diffusivities of $Li_{6.25}PS_{5.25}(BH_4)_{0.75}$ at 300 K and $Li_6POS_4(SH)$ at 400 K used in Fig. 1 of the paper.

”

Based on the statistical variance analysis, we added the estimated uncertainties of the room-temperature ionic conductivities of $Li_{6.25}PS_{5.25}(BH_4)_{0.75}$ and $Li_6POS_4(SH)$ in Fig. 1C in the revised paper:

In the caption of Fig. 1C, we added:

“...The uncertainties (in the parentheses) of the σ_{300K} for $Li_{6.25}PS_{5.25}(BH_4)_{0.75}$ and $Li_6POS_4(SH)$ are estimated based on the statistical variance analysis introduced in [37]....”

In these added new sections, two new references mentioned by the reviewer were cited in the revised paper as Ref. 37-38:

“37. He, X.; Zhu, Y.; Epstein, A.; Mo, Y., *Statistical variances of diffusional properties from ab initio molecular dynamics simulations. npj Computational Materials* 4: 18, 2018.

38. Liang, J. et al., *Site-Occupation-Tuned Superionic Li_xScCl_{3+x} Halide Solid Electrolytes for All-Solid-State Batteries, J. Am. Chem. Soc.* 142, 7012-7022, 2020.”

Reviewer’s comment 2:

2) The authors propose the design strategy of ALiSIC by incorporating small mono-anion clusters into a modern fast-ion framework. They show that the substitution of the mono-anion cluster (SH- or BH₄-) to lithium argyrodite (Li₆PS₅X) makes them ALiSIC with RT ionic conductivity ~0.1 S/cm and activation energies ~0.1 eV. I don’t think that this strategy can be applied to the known fast-ion frameworks such as LGPS, garnet-LLZO, NASICON, and perovskite-LLTO as they do not have halogen sites that can be substituted by the mono-anion cluster. More detailed discussion on the applicable fast-ion framework in addition to lithium argyrodite family should be provided to claim that this strategy is powerful.

Author’s response:

Following the reviewer’s advice, we have added a new paragraph in the Conclusion section to discuss the applicability of introducing mono-anion clusters into modern fast-ion frameworks besides the lithium argyrodite family. On page 27 of the revised paper, we added:

“Thirdly, the study shows that incorporating small mono-anion clusters into a modern fast-ion framework can serve as a powerful strategy to achieve lithium superionic conductors with high transport numbers. Mono-anion clusters, such as OH⁻, SH⁻, CN⁻ and BH₄⁻, have

similar ionic radii compared to the halogen group elements. Therefore, any ionic-conducting structures that can accommodate halogen site can be potential candidates for such a strategy. Indeed, we have seen this realized experimentally in Li/Na-rich antiperovskites $(\text{Li/Na})_{3-x}(\text{O/S})\text{X}$ ($\text{X} = \text{halogen or cluster}$) [41-42] and lithium argyrodites [43-44]. Recently discovered solid electrolyte systems with good properties, such as $\text{Li}_x\text{ScCl}_{3+x}$ [45], $\text{Na}_{3-x}\text{Y}_{1-x}\text{Zr}_x\text{Cl}_6$ [46] and $\text{Na}_{3-y}\text{PS}_{4-x}\text{Cl}_x$ [47], can also be subject to such a strategy....”

The following new references were cited as Ref. 41-47 in the revised paper:

“41. Hood, Z. D., Wang, H., Pandian, A. S., Keum, J. K. and Liang, C., *Li₂OHCl Crystalline Electrolyte for Stable Metallic Lithium Anodes*. *J. Am. Chem. Soc.* 138, 1768–1771, 2016.

42. Sun, Y. et al., *Rotational Cluster Anion Enabling Superionic Conductivity in Sodium-Rich Antiperovskite Na₃OBH₄*, *J. Am. Chem. Soc.* 141, 5640-5644, 2019.

43. Maughan, A. E.; Ha, Y.; Pekarek, R. T. and Schulze, C. M., *Lowering the Activation Barriers for Lithium-Ion Conductivity through Orientational Disorder in the Cyanide Argyrodite Li₆PS₅CN*, *Chem. Mater.* 33, 5127-5136, 2021.

44. Sakuda, A., Yamauchi, A., Yubuchi, S., Kitamura, N., Idemoto, Y., Hayashi, A. and Tatsumisago, M., *Mechanochemically Prepared Li₂S–P₂S₅–LiBH₄ Solid Electrolytes with an Argyrodite Structure*. *ACS Omega* 3, 5453-5458, 2018.

45. Liang, J. et al., *Site-Occupation-Tuned Superionic Li_xScCl_{3+x}Halide Solid Electrolytes for All-Solid-State Batteries*, *J. Am. Chem. Soc.* 142, 7012-7022, 2020.

46. Wu, E. A. et al., *A stable cathode-solid electrolyte composite for high-voltage, long-cycle-life solid-state sodium-ion batteries*, *Nature Communications* 12: 1256, 2021.

47. Feng, X. et al., *Studies of Functional Defects for Fast Na-Ion Conduction in Na_{3-y}PS_{4-x}Cl_x with a Combined Experimental and Computational Approach*, *Adv. Funct. Mater.* 1807951, 2019.”

Please also note that in the Introduction section of the paper, we emphasized the recent applications of the cluster-ions in various ionic conduction systems to promote high ionic conductivities:

“Most recently, superionic conductors (such as Li_3PS_4 , $\text{Li}_2\text{B}_{12}\text{H}_{12}$, $\text{NaCB}_9\text{H}_{10}$, LiNaSO_4 , Na_2SO_4 , Na_3PO_4 , LiBH_4 , Li_3SBF_4 , Na_3SBCl_4 , etc.) containing anion clusters (i.e., PS_4^{3-} , $\text{B}_{12}\text{H}_{12}^{2-}$, $\text{CB}_9\text{H}_{10}^-$, SO_4^{2-} , PO_4^{3-} , BH_4^- , BF_4^- , BCl_4^- , etc.) have become a fertile ground to search for high ionic conductivities [21-30]. An interesting feature of these ionic conductors is the so-called paddle-wheel effect characterized by strong correlation between the translational motion of the cation and the rotation of the cluster [21]. Under thermal excitation, non-spherical clusters can lead to a rotationally disordered phase, which can create low-energy pathways for ionic migrations and greatly enhance the ionic conductivity [10, 22-23, 31]. This is exemplified by the high Na-ion conductivity of 30 mS/cm of $\text{NaCB}_9\text{H}_{10}$ as it enters a rotational disordered phase at room temperature. Therefore, fully exploiting the rotational degrees of freedom and utilizing the paddle-wheel effect could be a viable way to achieve lithium conductors with near-ASIC metrics. However, other than $\text{NaCB}_9\text{H}_{10}$, most of the known cluster-containing ionic conductors can only transition to the rotational disordered phase at high temperatures [32-34]. It has been found that an amorphous structure with a low density and a relatively large volume to accommodate the cluster can support the paddle-wheel effect even at low temperatures (e.g., room temperature) [21, 31]. Yet, the rotational freedom of the clusters in such a system seems to be limited, as it is still not comparable to that of the high-temperature phase of $\text{Li}_2\text{B}_{12}\text{H}_{12}$ [21, 23]. On the other hand, one notices that the high charge states of the clusters (e.g., '-3' of PS_4^{3-} and '-2' of $\text{B}_{12}\text{H}_{12}^{2-}$) and their large moment of inertia may increase their local interactions and hinder their rotations under thermal excitation. Therefore, a combination of light mono-anion clusters with a chemically/structurally accommodating lattice framework could be the key to achieve exceptional lithium conductors that enjoy great rotational degrees of freedom of cluster at room temperature.”

Reviewer's comment 3:

3) In page 11, the statement, "... with small rotational barriers of SH⁻ and BH₄⁻ estimated to be about 30 and 10 meV, respectively.", was written. How to estimate the rotational barriers of the clusters?

Author's response:

Following the reviewer's advice, we have added in the revised paper our detailed method to estimate the rotational barriers of the clusters. On page 11 of the revised paper, we wrote:

"...Therefore, the orientationally disordered phase can be readily achieved in Li₆PO₄(SH) and Li_{6.25}PS_{5.25}(BH₄)_{0.75} with small rotational barriers of SH⁻ and BH₄⁻ estimated to be about 30 and 10 meV/atom (see Section II in SI), respectively...."

We added a new Section II in the Supplemental Information:

“

Section II: Estimation of the rotational barriers of BH₄⁻ and SH⁻ in the argyrodite-type structures. Supporting movies made from the trajectory data from the NEB calculations are given in the following link:

https://www.dropbox.com/sh/bzbp43x5hmftqxb/AACY4AKaITBb_u0rYjclOO7da?dl=0

The rotational barriers of BH₄⁻ and SH⁻ in their structures are estimated using nudged elastic band (NEB) method and selective dynamics. We use selective dynamics to fix the cluster under study to its designated angle, while allowing the other atoms in the supercell as well as the lattice parameters to fully relax during the NEB calculations. It is found that this method can characterize the expected rotation of the selected cluster without involving rotations of the other clusters in the supercell during the relaxation. On the other hand, the atoms in the supercell and the lattice parameters can still be relaxed during the calculation. This is shown in Movie S1-S3 made from the trajectories from the NEB calculations. For the case of SH⁻, one such cluster

in the supercell is rotated about its C_2 axis (viewed as a rigid rod) up to 180 degrees with a 30-degree step size. Within each step, four intermediate states are calculated by the NEB method. A total of 24 intermediate states are calculated between 0 – 180 degrees. The results are shown in Fig. S-II-1(A). The averaged rotational energy barrier per atom (two atoms are moving about the C_2 axis in the rotation) is 29 meV. For the case of BH_4^- , one such cluster in the supercell is rotated about one of its C_2 axes (as a rigid tetrahedron) up to 90 degrees; and is rotated about one of its C_3 axes up to 120 degrees in another study. A total of 12 intermediate states are calculated in the C_2 rotation. A total of 16 intermediate states are calculated in the C_3 rotation. The results are shown in Fig. S-II-1(B). It is found that the averaged rotational energy barrier per atom is 11 and 12 meV for C_2 rotation (with 4 H atoms moving) and C_3 rotation (with 3 H atoms moving), respectively.

Fig. S-II-1 Calculated energy profiles of rotation of the clusters. (A) Energy barriers (EB) of SH rotating as a rigid rod about its C_2 axis. (B) Energy barriers (EB) of BH_4^- as a rigid tetrahedron about its C_2 and C_3 axes, respectively. The small difference in the starting and ending states are due to the relaxation of the atoms and lattice parameters of the supercell.

”

Reviewer’s comment 4:

4) The authors separate cluster rotation types, thermally-excited rotation (‘active’ rotation), and the rotation interacted with Li-ion (‘responsive’ rotation). To confirm these effects,

they conducted NEB calculations with/without other atomic motions by using selective dynamics. But, the method to confirm rotation types, especially 'active' rotation, is confusing (Fig. 6). Please discuss this.

Author's response:

As pointed out by the reviewer, our original description and discussion about Fig. 6 with respect to the two types of cluster dynamics was not so clear. To make the discussion clearer and more concise, we rewrote the whole section related to Fig. 6 in the revised paper (on Page 20-25):

“Thus, besides the commonly-known paddle-wheel effect that only emphasizes the ionic diffusion promoted by the cluster rotation, a more complete picture emphasizing the interplay between the ion transport and the cluster dynamics (both rotational and translational) is formed. Here, the cluster dynamics can be categorized as ‘active’ (rotational and translational) dynamics due to the thermal excitation of the cluster, and ‘responsive’ dynamics due to the cluster’s reaction to a passing Li-ion. Given the great freedom of SH and BH₄ clusters in their corresponding systems, their (thermally-excited) ‘active’ dynamics and their ‘responsive’ dynamics to the nearby transporting Li-ion are always entangled. One needs to separate the impacts of these two kinds of dynamics for a better understanding. To achieve this, we devise a set of calculations on the distinctive Li-ion migration routes in each structure (Fig. 6), according to the routes of both intra- and inter-diffusions observed in the studied Events.

For the effect of the 'responsive' dynamics, we conduct two sets of calculations for the low-energy pathway along each route using the nudged elastic band (NEB) method. One is to fix all the atoms except the migrating Li-ion. The other is to only allow the Li-ion and its interacting cluster (or doped-sulfur for some cases in Li_{6.25}PS_{5.25}(BH₄)_{0.75}) to move, so that the cluster can relax and respond to the Li-ion movement during its migration. The difference between these two sets of calculations can show how the responsive dynamics of a cluster can facilitate (or hinder) the migration of the Li-ion. Meanwhile, the calculation can also show which migration route is energetically preferable.

For the effect of the 'active' dynamics of cluster, we manually rotate the cluster to different angles about its high-symmetry axes. Only the cluster under study is rotated, while the other ones in the supercell are fixed to their ground-state orientation. With each rotated angle of the cluster, we calculate the low-energy pathway for each migration route. All atoms except the migrating Li-ion are fixed during the calculation. This is to simulate the 'active' rotation of a cluster due to thermal excitation and see how the rotational angle can change the energy pathway of the Li-ion. Specifically, how such an 'active' rotation can change the relative energy between the starting and ending sites of the migration, as well as the energy extreme of the transition state along the pathway. Note that, due to the selective dynamics and the fixed volume of the cell during these NEB calculations, the obtained energy barriers are expected to be significantly overestimated. However, it is adequate for our purpose here to draw qualitative conclusions.

Fig. 6A shows the studied migration routes in the $\text{Li}_6\text{PO}_4(\text{SH})$ structure, where P1, P2 and P3 are three distinctive intra-diffusion routes while P4 and P5 are two distinctive inter-diffusion routes. The results for the effects of responsive dynamics of cluster are shown in Fig. 6B. These include the calculated energy barriers with fixed atoms except the migrating Li-ion (denoted with 'F') and the energy barriers allowing the interacting cluster (or the interacting doped sulfur) to relax (denoted with 'M'). It is found that, in all cases, allowing the cluster (or doped sulfur) to relax and respond to the moving Li-ion will greatly lower (up to 80%) the migration barrier, for both intra- and inter-diffusions, i.e., P1(F) vs. P1(M), P2(F) vs. P2(M), P4(F) vs. P4(M) and P5(F) vs. P5(M). Note that the intra-diffusion route P3 is away from any cluster. Therefore, its energy barrier is barely related to the cluster dynamics. The inset of Fig. 6B shows the trajectory of the typical responsive dynamics of a SH cluster to a passing Li-ion (in green). The above results show the cluster responding to the passing Li-ion via its rotational and translational dynamics which will always lower the energy barrier.

It is also found that the intra-diffusion routes of P1 and P2 show the lowest migration barriers (in the order of 10 meV) of all. P1 is characterized by the Li-ion interacting with one SH cluster (SH1) along its pathway, and P2 is characterized by the Li-ion migrating between two SH clusters (SH1 and SH2). Their low migration barriers explain why such intra-diffusions can be readily thermally-excited and why all the intra-

diffusions in the studied Events belong to the P1 and P2 types. For example, as shown in Fig. 4, Li(2)-SH(2) and Li(41)-SH(1) in Event 1 belong to P1; Li(19) migrating between SH(3) and SH(7) in Event 2 belongs to P2; and Li(27)-SH(3) belongs to Event 3. As expected, the calculated migration barriers of the inter-diffusion routes P4 and P5 are significantly greater than those of P1 and P2. This explains why only a few inter-diffusions are observed in the studied Events, with only Li(48) in Event 2 and Li(27) in Event 3 in Fig. 4. Both of these inter-diffusions are triggered by intra-diffusions. The above findings suggest that the 'billiard-ball' mechanism, featured by realizing long-range ionic conduction by the relay of local ionic diffusions, is the origin of the high ionic conductivity at low temperatures.

The results for the effects of active dynamics of cluster is shown in Fig. 6C. The SH cluster is rotated up to 300 degrees with the trajectory of its sulfur end shown in the inset of Fig. 6C. For each route, the relative energy difference (Diff) between the starting and the ending sites of the migration (in dashed line) as well as the energy extreme (Ext) along the route (in solid line) are calculated. It is found that the active rotation of a cluster can significantly change the pathway energy of Li-ion, with greatly lowered barrier at some angles and greatly increased barrier by the other, as shown by P1(Ext), P2(Ext), P4(Ext) and P5(Ext) in Fig. 6C. For example, when SH rotates to 125 degrees, the energy barrier of P1(Ext) is lowered by about 70%, while increased by about 80% when the cluster is further rotated to 230 degrees. The relative energy between the starting and ending sites of a route is also changed by the active rotation, as shown by P1(Diff), P2(Diff), P4(Diff) and P5(Diff) in Fig. 6C. These suggest that an active rotation of a cluster to certain angle can either trigger/promote or inhibit/suppress the ionic diffusion along certain route. Therefore, compared to the responsive dynamics of cluster which will always facilitate the Li-ion diffusion, the thermally-excited active rotation will introduce some randomness into the ionic-transport picture. Fig. 6C also shows that the effect of the active rotation is route-dependent. For example, the active rotation can change the pathway energies of P1(Ext) and P4(Ext) much more dramatically than it can change those of P2(Ext) and P5(Ext).

Fig. 6 Studies to understand the respective effects of the 'responsive' dynamics and the 'active' rotation of the clusters. (A) The distinctive migration routes in the structure of $\text{Li}_6\text{POS}_4(\text{SH})$. (B) The calculated migration barriers for different routes with and without the fixed dynamics of the cluster. The inset shows the rotational dynamics of a SH cluster responding to the movement of a Li-ion to lower the energy along the pathway. (C) Calculated energy difference (Diff) between the starting and the ending sites of a migration route as well as the energy extreme (Ext) of the transition state in the pathway vs. different rotational angles of the SH cluster. The inset shows the rotation trajectory of the sulfur end of the cluster. (D) The distinctive migration routes in the structure of $\text{Li}_{6.25}\text{PS}_{5.25}(\text{BH}_4)_{0.75}$. (E) The calculated migration barriers for different routes with and without the fixed dynamics of the cluster. The inset shows the rotational dynamics of a BH_4 cluster responding to the motion of a Li-ion to lower the energy along the pathway. (F) Calculated energy difference (Diff) between the starting and the ending sites of a migration route as well as the energy extreme (Ext) of the transition state in the pathway vs. different rotational angles of the BH_4 cluster about its C_2 and C_3 axes as shown in the inset.

Similar results are found for the responsive and active dynamics of BH_4 in $\text{Li}_{6.25}\text{PS}_{5.25}(\text{BH}_4)_{0.75}$ for the migration routes shown in Fig. 6D. The effect of the responsive dynamics of the BH_4 cluster is more pronounced than that of SH, as shown in Fig. 6E. Note that, for the P1 route, if we fix the boron atom of the cluster and only allow the hydrogen atoms to move, the lowered energy of P1 becomes less, as shown by the dotted line vs. the

solid line in red in Fig. 6E. This suggests that the translational dynamics of the cluster also contribute to the responsive dynamics of cluster significantly. The P2 route between the two BH₄ (1) and (2) clusters, and the P3 and P4 routes that are around the doped sulfur exhibit the flattest migration barriers. This explains why the P2-type intra-diffusions between two clusters are widely found in the studied Events, such as those between the B1 and B6 clusters in Event 4 as well as between the B4 and B6 clusters in Events 5 and 7 in Fig. 5.

In addition to their flatness, the energy barriers of both P3 and P4 will not be affected by any active rotations of the BH₄ (as demonstrated by P3 in Fig. 6F), since both P3 and P4 are unrelated to any BH₄ cluster as shown in Fig. 6D. This explains why all the major events identified for Li_{6.25}PS_{5.25}(BH₄)_{0.75} involve the doped sulfur, with Event 6 involving only the doped sulfur and no BH₄ cluster at all, as shown in Fig. 5. Note that there are only two doped sulfur sites out of eight BH₄ sites in the simulation cell. This suggests that the doped sulfur sites play a critical role for the ionic diffusion in Li_{6.25}PS_{5.25}(BH₄)_{0.75}. Indeed, the stoichiometric Li₆PS₅(BH₄) exhibits much lower ionic conductivity than those of Li_{6.25}PS_{5.25}(BH₄)_{0.75} and Li₆POS₄(SH) (Fig. 1C). This may be explained by its lack of the doped sulfur sites, leading to fewer Li-ion diffusion events as shown in Fig. S6 of Section IV in SI and being more susceptible to the adverse effects that could be brought by the active rotations of BH₄ due to excessive thermal excitation of the clusters found in Li₆PS₅(BH₄) (Fig. S10 of Section IV in SI).”

Reviewer’s comment 5:

5) Fig. 5: The Event numbers are wrong (1,2,3,4 should be 4,5,6,7)

Author’s response:

We have corrected this. Please refer to Fig. 5 on Page 16 of the revised paper.

Reviewer’s comment 6:

6) Fig. 6: In the plot of migration barriers, there are no explanations about abbreviations such as 'F', 'M', 'PM', and 'Ext'. Please demonstrate them.

Author's response:

In the revised paper, we have clearly defined, explained and demonstrated these annotations. On Page 21 of the revised paper, we wrote:

"...The results for the effects of responsive dynamics of cluster are shown in Fig. 6B. These include the calculated energy barriers with fixed atoms except the migrating Li-ion (denoted with 'F') and the energy barriers allowing the interacting cluster (or the interacting doped sulfur) to relax (denoted with 'M'). It is found that, in all cases, allowing the cluster (or doped sulfur) to relax and respond to the moving Li-ion will greatly lower (up to 80%) the migration barrier, for both intra- and inter-diffusions, i.e., $P1(F)$ vs. $P1(M)$, $P2(F)$ vs. $P2(M)$, $P4(F)$ vs. $P4(M)$ and $P5(F)$ vs. $P5(M)$"

On Page 22-23 of the revised paper, we wrote:

"...For each route, the relative energy difference (Diff) between the starting and the ending sites of the migration (in dashed line) as well as the energy extreme (Ext) along the route (in solid line) are calculated. It is found that the active rotation of a cluster can significantly change the pathway energy of Li-ion, with greatly lowered barrier at some angles and greatly increased barrier by the other, as shown by $P1(Ext)$, $P2(Ext)$, $P4(Ext)$ and $P5(Ext)$ in Fig. 6C. For example, when SH rotates to 125 degrees, the energy barrier of $P1(Ext)$ is lowered by about 70%, while increased by about 80% when the cluster is further rotated to 230 degrees. The relative energy between the starting and ending sites of a route is also changed by the active rotation, as shown by $P1(Diff)$, $P2(Diff)$, $P4(Diff)$ and $P5(Diff)$ in Fig. 6C. These suggest that an active rotation of a cluster to certain angle can either trigger/promote or inhibit/suppress the ionic diffusion. Therefore, compared to the responsive dynamics of cluster which will always facilitate the Li-ion diffusion, the thermally-excited active rotation will introduce some randomness into the ionic-transport picture. In addition, Fig. 6C shows that the effect of the active rotation is route-dependent. For example, the active rotation can change the pathway energies of $P1(Ext)$ and $P4(Ext)$ much more dramatically than it can change those of $P2(Ext)$ and $P5(Ext)$."

In the caption of Fig. 6, we added the meanings of ‘F’, ‘M’, ‘Diff’ and ‘Ext’:

“... (A) The distinctive migration routes in the structure of $\text{Li}_6\text{POS}_4(\text{SH})$. (B) The calculated migration barriers for different routes with (F) and without (M) the fixed dynamics of the cluster. The inset shows the rotational dynamics of a SH cluster responding to the movement of a Li-ion to lower the energy along the pathway. (C) Calculated energy difference (Diff) between the starting and the ending sites of a migration route as well as the energy extreme (Ext) of the transition state in the pathway vs. different rotational angles of the SH cluster. The inset shows the rotation trajectory of the sulfur end of the cluster....”

Reviewer’s comment 7:

7) In page 24, the statement, “... as shown in Fig. 6F, the energy barriers of both P4 and P3 will not be affected by the ‘active’ rotations ...” was written, but, in Fig. 6F, ‘P4’ doesn’t exist. Please check this.

Author’s response:

The P4 route is unrelated to any BH_4 cluster (as shown in Fig. 6D) and, therefore, is not included in Fig. 6F, since Fig. 6F is to discuss how the active rotation of the BH_4 cluster can influence the pathway energy of a route. However, P4 is related to a doped sulfur site (as shown in Fig. 6D). That is why we include it in Fig. 6E to show how its energy barrier will change due to the responsive dynamics of the doped sulfur. To clarify this, we added in the revised paper (on Page 25):

“... and the P3 and P4 routes that are around the doped sulfur exhibit the flattest migration barriers....”

“In addition to their flatness, the energy barriers of both P3 and P4 will not be affected by any active rotations of the BH_4 (as demonstrated by P3 in Fig. 6F), since both P3 and P4 are unrelated to any BH_4 cluster as shown in Fig. 6D....”

Reviewer #2 (Remarks to the Author):

In their revision, the authors made substantial changes. In fact, only 4 (!) of the first 25 pages of the original manuscript appear to be the same, all other pages have been completely rewritten. I have to say that this is quite unusual even for a major revision.

However, the manuscript has indeed improved in the revision. Especially the discussion of prior work is now much more thorough and better organized. Nevertheless, the analysis of the Li diffusion mechanism, although completely redone, remains overly complex and could be shortened substantially. This is also the case for the conclusion section, which could be shortened by ~50% and should focus on the take-home messages.

Reviewer's comment 1:

I suggest the authors revise the manuscript once again, this time for conciseness. I recommend to focus on the most important questions, for example, to work out clearly: (i) What is the origin of fast long-range Li conduction in the materials? (ii) In which way is the mechanism similar to or different from previous reports (e.g., paddle-wheel mechanism)? (iii) Can the insight be used to formulate a general design criterion? These points are still not clear.

Author's response:

We thank the reviewer for this advice. Accordingly, we rewrote the sections on Page 18-25 that are related to the dynamic correlation between the cluster-ion and the Li-ion. Please refer to these discussions in the revised paper. We also rewrote the discussions in the Conclusion section, focusing on the key points and making it concise. Regarding the three questions mentioned by the reviewer, we added the following discussion in the revised paper (Page 25-27):

“Several key conclusions can be drawn and further discussed from the above. First, the origin of the high ionic conductivity of the materials at low temperatures is attributed to

the 'billiard-ball' mechanism. It was believed that local ionic diffusions cannot contribute to the long-range ionic conduction of a material. However, the 'billiard-ball' mechanism shows that long-range ionic conduction can be effectively realized by a relay of local diffusions across the lattice sustained by Li-Li repulsions. Such mechanism is also found for the ionic conduction of the nonstoichiometric lithium argyrodite $\text{Li}_{6.25}\text{PS}_{5.25}\text{Cl}_{0.75}$, as shown in Fig. S11 of Section IV in SI. To sustain the 'billiard-ball' mechanism, it is important for the Li-Li repulsion to overcome the pathway barrier. This can be supported by four factors in the studied systems: (1) the low energy barriers of intra-diffusions in a sulfur block; (2) the responsive (rotational and translational) dynamics of cluster to lower the energy barriers for local diffusions as demonstrated in $\text{Li}_{6.25}\text{PS}_{5.25}(\text{BH}_4)_{0.75}$ and $\text{Li}_6\text{POS}_4(\text{SH})$; (3) doped sulfur sites as weak interacting conduit to lower the energy barriers for local diffusions as demonstrated in $\text{Li}_{6.25}\text{PS}_{5.25}(\text{BH}_4)_{0.75}$ and $\text{Li}_{6.25}\text{PS}_{5.25}\text{Cl}_{0.75}$; (4) Li-ion excess to enhance the Li-Li repulsive interaction in the nonstoichiometric configurations $\text{Li}_{6.25}\text{PS}_{5.25}(\text{BH}_4)_{0.75}$ and $\text{Li}_{6.25}\text{PS}_{5.25}\text{Cl}_{0.75}$. The 'billiard-ball' mechanism is especially useful to achieve high ionic conductivity at low temperatures and should be considered in the future design of solid electrolytes. Other uncovered mechanisms in this study, including the 'revolving-door' mechanism and the 'docking-undocking' mechanism with significant time scale (over tens of pico-seconds) can be also helpful for our understanding of ionic diffusion process in solids.

Secondly, besides the paddle-wheel effect that merely emphasizes on the Li-ion migration promoted by the cluster rotation, the current study shows a more complete picture of the dynamic correlation between the cluster-ions and Li-ions with the following new aspects: (1) The dynamics of cluster can be categorized into 'responsive' and 'active' dynamics. The former is characterized by accommodating the Li-ion movement and will always lower the migration barrier. The latter is characterized by active rotation of clusters due to thermal excitation, which, depending on the migration route, may facilitate or inhibit the ionic diffusion. (2) Both translational and rotational degrees of freedom (not just the rotation) of the cluster can contribute significantly to its dynamic correlation with the Li-ion."

Please note that we also added the following discussions related to the ‘Paddle-wheel’ effect on Page 18-20 of the revised paper:

“Fourth, the paddle-wheel effect found in $\text{Li}_6\text{POS}_4(\text{SH})$ and, especially in $\text{Li}_{6.25}\text{PS}_{5.25}(\text{BH}_4)_{0.75}$, are not quite the same as that in the lithium conductors with heavy polyanions (e.g., Li_3PS_4 in Ref. 21). One content of the paddle-wheel effect is that the rotation of a cluster can ‘drag/pull’ its nearby Li-ion to move along. This suggests very strong interaction between the cluster and the Li-ion, which is not likely to be favored by a mono-anion with small charge states on its vertices (e.g., BH_4^-) as compared to polyanions (e.g., PS_4^{3-}). It also suggests that the motion of the Li-ion will not significantly influence the dynamics of the cluster, which is not favored by a light and small cluster (e.g., SH^- or BH_4^-) as compared to a heavy and large cluster such as PS_4^{3-} or $\text{B}_{12}\text{H}_{12}^{2-}$. This suggests that a pronounced rotation of the cluster should lead or precede the related motion of the Li-ion. However, it would be difficult to have a clear picture of this in a complex system, when pronounced rotational dynamics can be readily thermally-excited throughout the time (as in the current cases of SH^- and BH_4^-) and the Li-ion is interacting with multiple clusters simultaneously. In the studied materials, it is hard to find a clear case in which the Li-ion displacement is initiated by a rotation of the cluster, as shown by the SH and BH_4 dynamics in Fig. 4 and 5. Rather, pronounced rotational dynamics are found after or, in some cases, during the displacement of the Li-ion. This suggests that the rotational dynamics of the clusters are more of a response to the motion of Li-ion. In addition, there are strong thermally-excited (rotational and translational) cluster dynamics in these materials, especially in $\text{Li}_{6.25}\text{PS}_{5.25}(\text{BH}_4)_{0.75}$, which significantly fazes the correlation, as shown by the big fluctuations in the rotational dynamics of SH and BH_4 clusters in Fig. 4 and 5, respectively. It is also found that the correlation between the Li-ion displacement and the cluster dynamics is more pronounced in $\text{Li}_6\text{POS}_4(\text{SH})$ than in $\text{Li}_{6.25}\text{PS}_{5.25}(\text{BH}_4)_{0.75}$, as shown by the rotational profile of the cluster in resemblance to that of the Li-ion displacement (e.g., $\text{Li}(48)\text{-SH}(8)$ in Fig. 4D, $\text{Li}(11)\text{-SH}(4,8)$ in Fig. 4E, $\text{Li}(19)\text{-SH}(3)$ in Fig. 4F, $\text{Li}(27)\text{-SH}(3)$ in Fig. 4G, $\text{Li}(17)\text{-SH}(3)$ in Fig. 4H and $\text{Li}(13)\text{-SH}(7)$ in Fig. 4I; also refer to Section III of SI). This is due to the fact that SH^- has a higher (Bader) charge state of its S (-0.9e), a much heavier weight and significantly less rotational

freedom under thermal excitation (judged by its angular decorrelation rate in Fig. S5 of Section IV of SI) than those of BH_4^- .

Thus, besides the commonly-known paddle-wheel effect that only emphasizes on the ionic diffusion promoted by the cluster rotation, a more complete picture emphasizing on the interplay between the ion transport and the cluster dynamics (both rotational and translational) is formed. Here, the cluster dynamics can be categorized as ‘active’ (rotational and translational) dynamics due to the thermal excitation of the cluster, and ‘responsive’ dynamics due to the cluster’s reaction to a passing Li-ion. Given the great freedom of SH and BH_4^- clusters in their corresponding systems, their (thermally-excited) ‘active’ dynamics and their ‘responsive’ dynamics to the nearby transporting Li-ion are always entangled. One needs to separate the impacts of these two kinds of dynamics for a better understanding. To achieve this, we devise a set of calculations on the distinctive Li-ion migration routes in each structure (Fig. 6), according to the routes of both intra- and inter-diffusions observed in the studied Events.”

Reviewer’s comment 2:

An important novel insight is (in my opinion) the impact of Li repulsion on the triggering of long-range diffusion (i.e., the "billiard ball mechanism"). Why is this mechanism relevant only for cluster ions? Many Li-ion conductors exhibit fast local Li diffusion that does not contribute to overall Li conductivity. Should the mechanism not generally be of importance for such materials?

Author’s response:

According to our current study, the Li-Li repulsion on triggering the ionic diffusions relies on the low energy barrier of the pathway. In the revised paper, we have added the following discussion (on Page 26):

“...It was believed that local ionic diffusions cannot contribute to the long-range ionic conduction of a material. However, the ‘billiard-ball’ mechanism shows that long-range ionic conduction can be effectively realized by a relay of local diffusions across the lattice

sustained by Li-Li repulsions. Such mechanism is also found for the ionic conduction of the nonstoichiometric lithium argyrodite $\text{Li}_{6.25}\text{PS}_{5.25}\text{Cl}_{0.75}$, as shown in Fig. S11 of Section IV in SI. To sustain the 'billiard-ball' mechanism, it is important for the Li-Li repulsion to overcome the pathway barrier. This can be supported by four factors in the studied systems: (1) the low energy barriers of intra-diffusions in a sulfur block; (2) the responsive (rotational and translational) dynamics of cluster to lower the energy barriers for local diffusions as demonstrated in $\text{Li}_{6.25}\text{PS}_{5.25}(\text{BH}_4)_{0.75}$ and $\text{Li}_6\text{POS}_4(\text{SH})$; (3) doped sulfur sites as weak interacting conduit to lower the energy barriers for local diffusions as demonstrated in $\text{Li}_{6.25}\text{PS}_{5.25}(\text{BH}_4)_{0.75}$ and $\text{Li}_{6.25}\text{PS}_{5.25}\text{Cl}_{0.75}$; (4) Li-ion excess to enhance the Li-Li repulsive interaction in the nonstoichiometric configurations $\text{Li}_{6.25}\text{PS}_{5.25}(\text{BH}_4)_{0.75}$ and $\text{Li}_{6.25}\text{PS}_{5.25}\text{Cl}_{0.75}$. The 'billiard-ball' mechanism is especially useful to achieve high ionic conductivity at low temperatures and should be considered in the future design of solid electrolytes. Other uncovered mechanisms in this study, including the 'revolving-door' mechanism and the 'docking-undocking' mechanism with significant time scale (over tens of pico-seconds) can be also helpful for our understanding of ionic diffusion process in solids.”

Please also refer to the results and discussions on Page 20-25 that support the above conclusions. In addition, to demonstrate the ‘billiard-ball’ mechanism in a non-cluster-containing system, we have added the ‘Event’ analysis on the nonstoichiometric lithium argyrodite $\text{Li}_{6.25}\text{PS}_{5.25}\text{Cl}_{0.75}$, as in the new Fig. S11 of Section IV in the Supplemental Information:

“

Fig. S11 Three combined events found for the nonstoichiometric lithium argyrodite $\text{Li}_{6.25}\text{PS}_{5.25}\text{Cl}_{0.75}$ at a simulation temperature of 500 K. $S(d)$ represents the doped sulfur site in structure. The Li-ion trajectories are in dark green. All three major events are found happening around the doped sulfur site. Event 1 involves the diffusion of Li(15) that originally bound to the doped sulfur to the nearby sulfur block (SLi_6), triggering a set of local diffusion in the block, according to the ‘billiard-ball’ mechanism as observed in the cluster-containing ALiSIC . Event 2 again shows a Li-ion Li(20), originally bound to the doped sulfur site, diffuses into a nearby sulfur block and triggers a set of intra-diffusion according to the billiard-ball mechanism. Event 3 is a typical ‘revolving-door’ mechanism featuring only local intra-diffusions in the same sulfur block.

”

Reviewer #3 (Remarks to the Author):

Authors have carried out an extensive revision, and the paper is almost entirely rewritten.
I don't have any other comments, at least on my end.

Author's response:

We thank the reviewer for the positive comment.

REVIEWERS' COMMENTS

Reviewer #1 (Remarks to the Author):

The authors have largely responded to the reviewers' comments and the revised manuscript has been improved with further analyses. I don't have any other comments.